# A molnupiravir-associated mutational signature in global SARS-CoV-2 genomes

Theo Sanderson[1 ✉], Ryan Hisner[2], I'ah Donovan-Banfield[3,4], Hassan Hartman[5], Alessandra Løchen[5], Thomas P. Peacock[6,7] & Christopher Ruis[8,9,10,11 ✉]

Molnupiravir, an antiviral medication widely used against severe acute respiratory syndrome coronavirus 2 (SARS-CoV-2), acts by inducing mutations in the virus genome during replication. Most random mutations are likely to be deleterious to the virus and many will be lethal; thus, molnupiravir-induced elevated mutation rates reduce viral load[1,2]. However, if some patients treated with molnupiravir do not fully clear the SARS-CoV-2 infections, there could be the potential for onward transmission of molnupiravir-mutated viruses. Here we show that SARS-CoV-2 sequencing databases contain extensive evidence of molnupiravir mutagenesis. Using a systematic approach, we find that a specific class of long phylogenetic branches, distinguished by a high proportion of G-to-A and C-to-T mutations, are found almost exclusively in sequences from 2022, after the introduction of molnupiravir treatment, and in countries and age groups with widespread use of the drug. We identify a mutational spectrum, with preferred nucleotide contexts, from viruses in patients known to have been treated with molnupiravir and show that its signature matches that seen in these long branches, in some cases with onward transmission of molnupiravir-derived lineages. Finally, we analyse treatment records to confirm a direct association between these high G-to-A branches and the use of molnupiravir.

Molnupiravir is an antiviral drug, licensed in some countries for the treatment of coronavirus disease 2019 (COVID-19). In the body, molnupiravir is ultimately converted into a nucleotide analogue, molnupiravir triphosphate (MTP; also known as β-D-*N*-hydroxycytidine triphosphate)[3]. MTP is incorporated into RNA during strand synthesis by viral RNA-dependent RNA polymerases, where it can result in errors of sequence fidelity during viral genome replication. These errors result in many viral progeny that are non-viable and thus reduce the virus's effective rate of growth. Molnupiravir reduced viral replication during 24 h by 880-fold in vitro, and reduced viral load both in animal models[1] and in patients sampled on the final day of treatment[2]. Molnupiravir initially showed some limited efficacy as a treatment for COVID-19 (refs. 3,4); subsequently, the much larger PANORAMIC trial found that treatment did not reduce hospitalization or death rates in a group of largely vaccinated high-risk individuals[2]. As one of the first orally bioavailable antivirals on the market, molnupiravir was widely adopted in many countries. However, recent trial results and the approval of more efficacious antivirals have since led to several countries recommending against its use[5–7], while longstanding concerns have been raised about potential mutagenic activity in host cells[8].

MTP is incorporated into nascent RNA primarily by acting as an analogue of cytosine (C), pairing opposite guanine (G) bases (Fig. 1a). However, once incorporated, the molnupiravir (M) base can transition into an alternative tautomeric form that resembles uracil (U) instead. This means that in the next round of strand synthesis, to give the positive-sense SARS-CoV-2 genome, the tautomeric M base pairs with adenine (A), resulting in a G-to-A mutation (Fig. 1b). These G-to-A mutations arise from incorporation of molnupiravir into the negative-sense genome. Incorporation of MTP can also occur during the synthesis of the positive-sense genome: in this scenario, an initial positive-sense C correctly pairs with a G during negative-sense synthesis, but this G then pairs with an M base during positive-sense synthesis. In the next round of replication, this M can then pair with A, which results in a U in the final positive-sense genome, with the overall process producing a C-to-U mutation (Fig. 1c). The free nucleotide MTP is less prone to tautomerization to the oxime form than when incorporated into RNA, and so this directionality of mutations is the most likely[9]. However, it is also possible for some MTP to bind, in place of U, to A bases and undergo the above processes in reverse, causing A-to-G and U-to-C mutations (Fig. 1c).

It has been suggested that many major SARS-CoV-2 variants emerged from long-term chronic infections. This model explains several peculiarities of variants, such as a general lack of genetic intermediates, rooting with much older sequences, long phylogenetic branch lengths and the level of convergent evolution with known chronic infections[10–13]. During the approval process for molnupiravir, concerns were raised about its potential to increase the rate of evolution of variants of

[1]Francis Crick Institute, London, UK. [2]Department of Bioinformatics, University of Cape Town, Cape Town, South Africa. [3]Department of Infection Biology and Microbiomes, Institute of Infection, Veterinary and Ecological Sciences, University of Liverpool, Liverpool, UK. [4]Health Protection Research Unit in Emerging and Zoonotic Infections, National Institute for Health and Care Research, Liverpool, UK. [5]UK Health Security Agency, London, UK. [6]Department of Infectious Disease, Imperial College London, London, UK. [7]The Pirbright Institute, Pirbright, UK. [8]Molecular Immunity Unit, University of Cambridge Department of Medicine, Medical Research Council-Laboratory of Molecular Biology, Cambridge, UK. [9]Department of Veterinary Medicine, University of Cambridge, Cambridge, UK. [10]Cambridge Centre for AI in Medicine, University of Cambridge, Cambridge, UK. [11]Victor Phillip Dahdaleh Heart & Lung Research Institute, University of Cambridge, Cambridge, UK. ✉e-mail: theo.sanderson@crick.ac.uk; cr628@cam.ac.uk

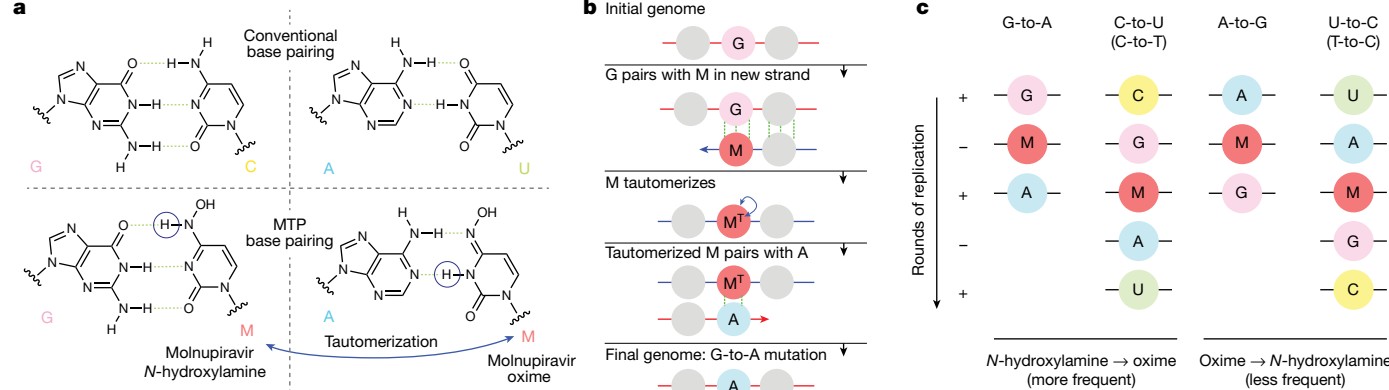

**Fig. 1 | Molnupiravir induces mutations by acting as a nucleotide analogue with multiple tautomeric forms that pair preferentially with different nucleotides. a**, Molnupiravir triphosphate can assume multiple tautomeric forms that resemble different nucleotides. The *N*-hydroxylamine form resembles cytosine (C); like cytosine, it can pair with guanine (G) while the oxime form more closely resembles uracil (U) and thus can pair with adenine (A) (figure adapted in part from Malone and Campbell[45]). **b**, In the most common scenario, molnupiravir (M) is incorporated in the *N*-hydroxylamine form opposite a G nucleotide. It can then tautomerize into the oxime form, which can then pair to an A in subsequent replication, creating a G-to-A mutation. **c**, Molnupiravir can result in four different mutation types. In the

first column, a G-to-A mutation is created by M incorporation opposite a positive-sense G, which can then pair with an A in the next positive-sense synthesis. In the second column, the positive-sense genome contains a C, which results in a G in the negative-sense genome. This G can then undergo the same G-to-A mutation, creating a negative-sense A that finally results in a U in the positive-sense genome, meaning that the entire process results in a C-to-U mutation. Although the biases of tautomeric forms for the free and incorporated MTP nucleotides favour this directionality of mutations, with M incorporated in the *N*-hydroxylamine form and then transitioning to the oxime form, the reverse can also occur: this results in A-to-G and U-to-C mutations.

concern[14]. In response, it was noted that no infectious virus had been isolated at or beyond day five of molnupiravir treatment, and that mutations recovered following treatment were random with no evidence of selection-based bias[15].

During analysis of divergent SARS-CoV-2 sequences, signs of molnupiravir-driven mutagenesis have been noted[16], including indications of possible transmission. Therefore, we aimed to characterize the mutational profile of molnupiravir and examine the extent to which this signature appeared in global sequencing databases.

## Emergence of a new mutational signature

To establish the mutational profile induced by molnupiravir, we analysed published longitudinal genomic time series that included both untreated patients and patients treated with molnupiravir[17,18], and compared them against a typical SARS-CoV-2 mutational spectrum[19]. In agreement with previous findings, we found that molnupiravir treatment led to a substantial increase (8×, 95% confidence interval = 2.9–16.0) in the rate of mutations and that this increase was highly specific to transition mutations (Fig. 2a), especially to G-to-A and C-to-T mutations (hereon we use 'T' rather than 'U' as in the sequences). While C-to-T mutations are relatively common overall in SARS-CoV-2 evolution[19–21], G-to-A mutations occur much less frequently; therefore, an elevated G-to-A proportion was especially predictive of molnupiravir treatment (Fig. 2b). We looked for evidence of such a signal in global sequencing databases by analysing a mutation-annotated tree derived from McBroome et al.[22], containing more than 15 million SARS-CoV-2 sequences from the Global Initiative on Sharing All Influenza Data (GISAID)[23] and the International Nucleotide Sequence Database Collaboration (INSDC) databases[24]. For each branch of the tree, we counted the number of each substitution class (for example, A-to-T, A-to-G). Filtering this tree to branches involving at least 20 substitutions and plotting the proportion of substitution types revealed a region of this space with higher G-to-A and almost exclusively transition substitutions, which only contained branches of sequences sampled since 2022 (Fig. 2c), suggesting that some change (either biological or technical) had resulted in a new mutational pressure, with mutation classes consistent with those in patients known to be treated with molnupiravir.

We created a criterion for branches of interest, which we refer to as 'high G-to-A' branches: we selected branches involving at least ten substitutions, of which more than 25% were G-to-A, more than 20% were C-to-T and more than 90% were transitions. Simulations predicted that this criterion would have a sensitivity of 63% and a specificity of 98.6% for branches involving 13 substitutions (Methods). Branches satisfying the high G-to-A conditions were almost all sampled after the roll-out of molnupiravir in late 2021 and early 2022 (Fig. 2d and Extended Data Fig. 1). Branches were predominantly sampled from a small number of countries, which could not be explained by differences in sequencing efforts (Fig. 2e,f and Extended Data Table 1).

Many countries that exhibited a high proportion of high G-to-A branches used molnupiravir: more than 380,000 prescriptions occurred in Australia by the end of 2022 (ref. 25). More than 30,000 occurred in the UK in the same period[2,26], more than 240,000 in the USA within the early months of 2022 (ref. 27) and more than 600,000 in Japan by October 2022 (ref. 28). Countries with high levels of total sequencing but a low number of G-to-A branches (Canada and France; Fig. 2e,f) have not authorized the prescription of molnupiravir[29,30]. Age metadata from the USA showed a significant bias towards samples from older patients for these high G-to-A branches, compared to control branches with similar numbers of mutations but without filtering on substitution type (Fig. 2g). Where age data were available in Australia, they also suggested that high G-to-A branches primarily occurred in an aged population. This is consistent with the prioritized use of molnupiravir to treat older individuals, who are at greater risk from severe infection, in these countries. In Australia, molnupiravir was preplaced in aged care facilities; it was recommended that it should be considered for all residents testing positive for COVID-19 aged 70 years or older, with or without symptoms[31].

## Mutation contexts support molnupiravir link

To further probe the link between high G-to-A branches and molnupiravir, we used mutation spectrum analysis, which considers both the types of mutations and the genomic context in which they occur (Extended Data Fig. 4). The spectrum we identified for branches selected by the high G-to-A criteria was, as expected,

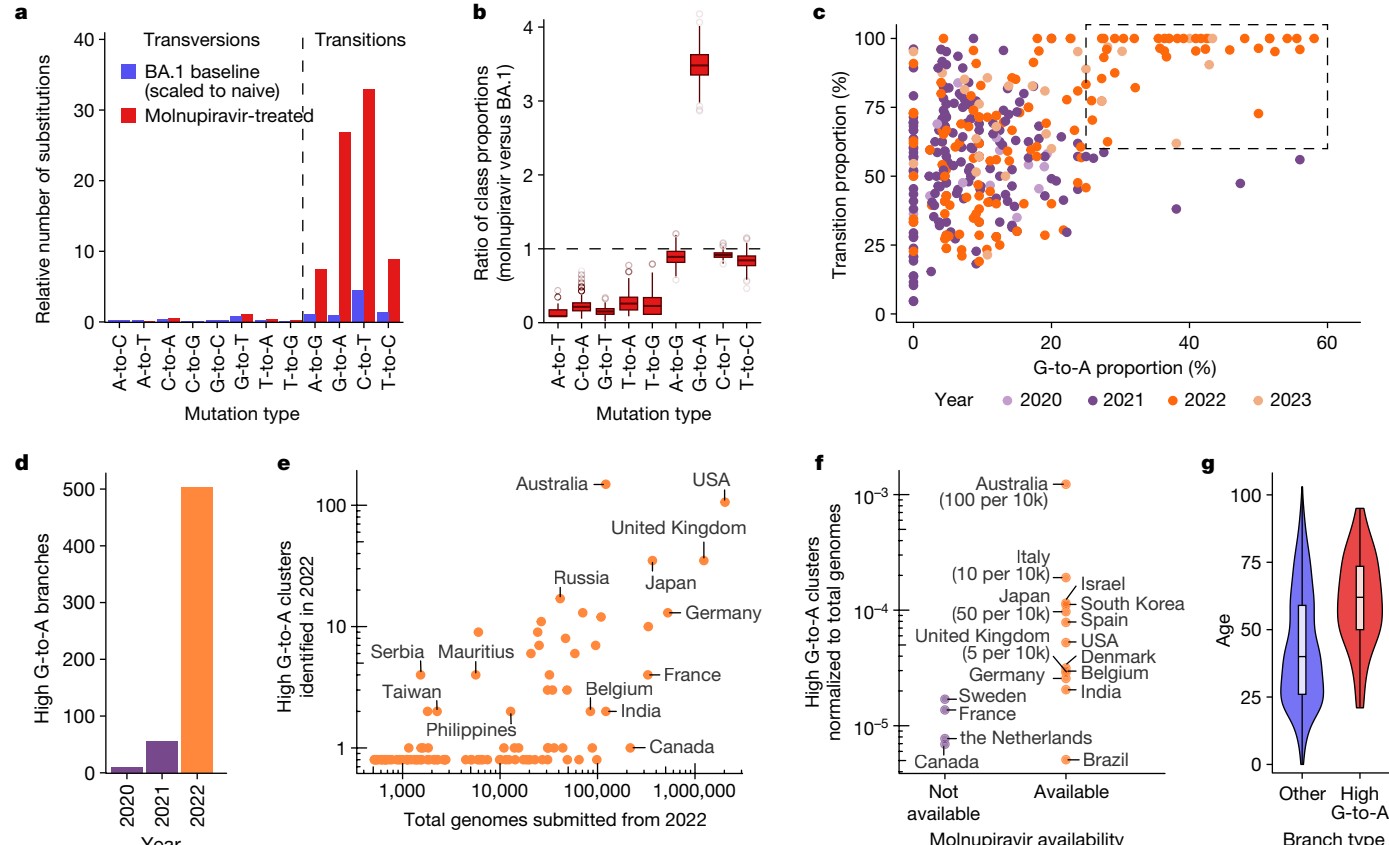

**Fig. 2 | A molnupiravir-associated mutational signature with high G-to-A and high transition ratio emerged in 2022 in some countries in global sequencing databases. a**, Comparison of the relative rate of different classes of mutations in typical BA.1 mutations versus those with molnupiravir treatment (molnupiravir data from Alteri et al.[17] and naive data from Ruis et al.[19], scaled to total mutations in naive individuals from Alteri et al.[17]; Methods) confirms an elevated rate of transitions, and particularly C-to-T and G-to-A mutations. **b**, Differences in the proportion of mutations of different mutation classes in individuals treated with molnupiravir (Alteri et al.[17]) versus typical BA.1 mutations (Ruis et al.[19]) highlight elevated G-to-A proportion as especially indicative of molnupiravir. These are ratios of proportions, so the apparent reduction in transversions does not require an absolute decrease in the number of transversions but can instead be caused by the increased number of transitions. The box plots depict variation over 1,000 bootstrap resamplings, with boxes showing the 25th, 50th and 75th percentiles, and the whiskers having a length 1.5× the interquartile range. **c**, A scatter plot where each point is a branch with more than 20 mutations, positioned according to the proportion of these mutations that are G-to-A (*x* axis) or transitions (*y* axis), reveals a space

with elevated G-to-A and transition rate that occurs only with the roll-out of molnupiravir in 2022. **d**, A change at the same time point is seen when plotting the number of nodes with more than ten mutations and with G-to-A proportion greater than 25%, C-to-T proportion greater than 20% and transition proportion greater than 90%. **e**, Plotting the number of high G-to-A nodes identified in 2022 against the number of total genomes for each country revealed considerable variation. **f**, Countries confirmed to have made molnupiravir available had more high G-to-A nodes than countries that did not. The numbers in brackets represent the number of courses of molnupiravir supplied, normalized to population. *P* = 0.02 for a log-transformed, two-sided *t*-test. **g**, Age distribution for US nodes, partitioned according to whether they satisfy the high G-to-A criteria (*P* < 1 × 10[−10], two-sided *t*-test). Age metadata are missing for some samples, probably non-randomly. Dataset sizes are *n* = 106 for the high G-to-A branches and *n* = 2,472 for the other branches. Where a node had many descendants of different ages, age was assigned using a basic heuristic, as described in the Methods. The box plot depicts the minimum, maximum, and the 25th, 50th and 75th percentiles.

dominated by transition mutations, with G-to-A and C-to-T directionality more common than A-to-G and T-to-C (Fig. 3a). We also calculated spectra both from patients known to be treated with molnupiravir[17,18] (Fig. 3b) and from general SARS-CoV-2 evolution[19] (Fig. 3c).

There was a strong match between the spectrum of known molnupiravir sequences and that of high G-to-A branches, both in terms of the proportions of different mutation classes and of the context preferences within each mutation class (Fig. 3 and Extended Data Fig. 5a). For C-to-T and G-to-A mutations, a comparison of context preferences gave cosine similarities of 0.988 and 0.965, respectively (Fig. 3d). Similar results were seen when comparing to a spectrum calculated from a separate second dataset from a clinical trial of molnupiravir (Extended Data Fig. 5b)[18]. The contextual patterns seen in long branches did not correlate with those during typical SARS-CoV-2 evolution (Extended Data Fig. 5c). In high G-to-A branches, G-to-A mutations occurred most commonly in

TGT and TGC contexts, which could represent a preference for molnupiravir binding adjacent to particular surrounding nucleotides, a preference of the viral RNA-dependent RNA polymerase to incorporate molnupiravir adjacent to specific nucleotides, or a context-specific effect of the viral proofreading machinery. These correlations between spectra from high G-to-A branches and individuals known to have been treated with molnupiravir strongly support a shared mutational driver, and therefore that the high G-to-A branches are driven by molnupiravir treatment.

Incorporation of molnupiravir during negative strand synthesis results in G-to-A mutations in the virus sequence, while incorporation during positive strand synthesis manifests as C-to-T mutations in the virus sequence after a second round of replication (Fig. 1c). Consistent with this, we observed a strong similarity between the mutational biases in equivalent surrounding contexts within C-to-T and G-to-A mutations when comparing reverse complement triplets with,

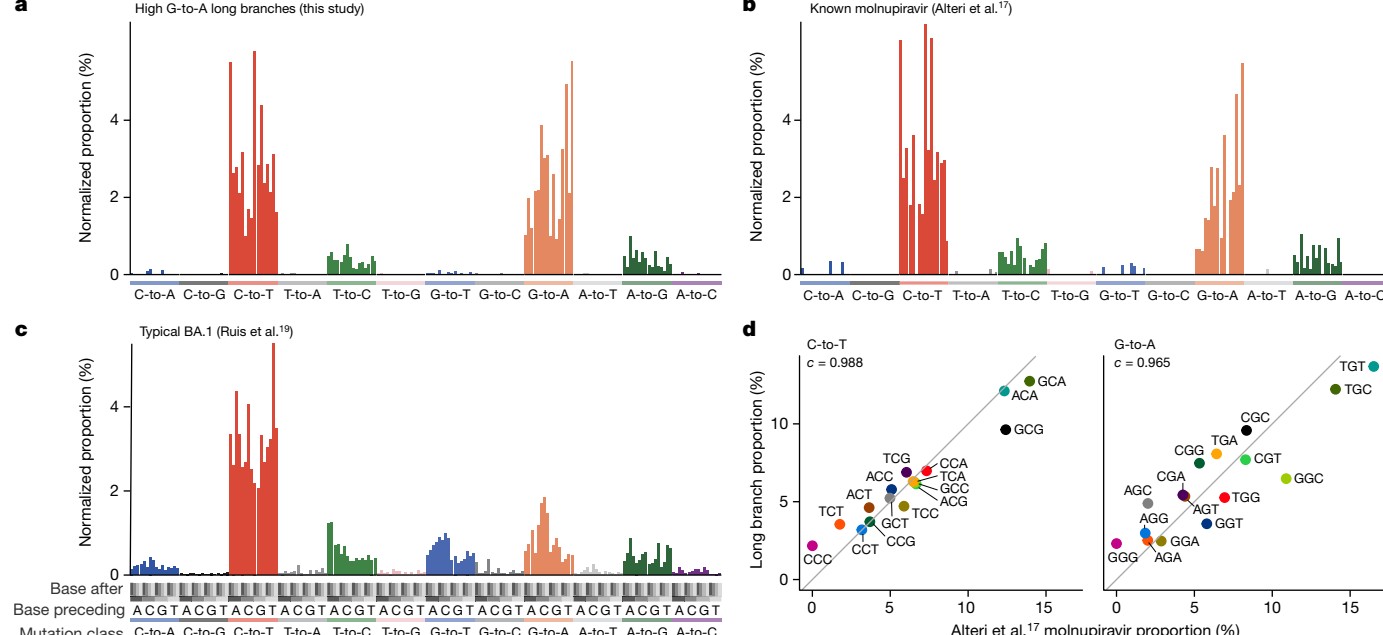

**Fig. 3 | Mutation spectrum analysis supports high G-to-A branches being driven by molnupiravir. a–c**, Single-base substitution mutation spectra for high G-to-A branches (**a**), individuals known to have been treated with molnupiravir (**b**) and typical BA.1 spectra (**c**). Each individual bar represents a particular type of mutation in a particular trinucleotide context (Extended Data Fig. 4). Bars are grouped and coloured according to the class of mutation. Within each coloured group, bars are grouped into four groups according to the nucleotide preceding the mutated residue; then, each of these groups contains four bars according to the nucleotide following the mutated residue.

The number of mutations has been normalized to the number of times the trinucleotide occurs in the reference genome, and then normalized so that the entire spectrum sums to 1. **d**, High correlations between spectra from Alteri et al.[17] from patients known to have been treated with molnupiravir, and the spectra from high G-to-A branches identified in this study. Each point represents the normalized proportion of a particular trinucleotide context. Points are coloured so that a context for C-to-T mutations has the same colour as its reverse complement in G-to-A. The values denoted by $c$ are cosine similarity scores.

for example, a G-to-A mutation in the TGC context on one strand being equivalent to a C-to-T mutation in the GCA context on the other strand (cosine similarity = 0.955; Extended Data Fig. 5d).

## Transmission clusters and mutation rates

Although most of the long high G-to-A branches identified had just a single descendant tip sequence in sequencing databases, in some cases we could see that branches had given rise to clusters with a significant number of descendant sequences. For example, a cluster in Australia in August 2022 involved 20 tip sequences, with distinct age metadata confirming they derived from multiple individuals (Fig. 4a). This cluster involved 25 substitutions in the main branch, of which all were transitions, with 44% C-to-T and 36% G-to-A. Closely related outgroups date from July 2022, suggesting that these mutations emerged in a period of 1–2 months. At the typical rate of SARS-CoV-2 evolution, this number of mutations would take years to acquire in an unsampled population with typical dynamics[32]. Overall in the dataset, we observed a systematic accelerated evolutionary rate in high G-to-A branches ($P < 0.001$, two-sided $t$-test), which is consistent with the action of a mutagenic drug.

There are many further examples of high G-to-A branches with multiple descendant sequences, including sequence clusters in the United Kingdom, Japan, USA, New Zealand, Slovakia, Denmark, South Korea and Vietnam (Fig. 4b and Supplementary Information).

During the construction of the daily updated mutation-annotated tree[22], samples highly divergent from the existing tree were excluded. This is a necessary step given the technical errors in some SARS-CoV-2 sequencing data, but means that highly divergent molnupiravir-induced sequences might have been missed from this analysis. To search for excluded sequences with a molnupiravir-like pattern of mutations, we processed a full sequence dataset with Nextclade and calculated the proportion of each of the mutation classes among the private mutations (Methods) each sequence carried. This analysis allowed the identification of further mutational events, including some involving up to 130 substitutions (Fig. 4c and Extended Data Fig. 2), with the same signature of elevated G-to-A mutation rates and almost exclusively transition substitutions. The cases we identified with these very high numbers of mutations predominantly involved single sequences and could represent sequences resulting from chronically infected individuals treated with multiple courses of molnupiravir. We verified that nucleotide contexts of the transition mutations observed within the sequence in Fig. 4c were much more likely under the molnupiravir spectrum than the typical BA.1 spectrum (Bayes factor > $10^{10}$).

## Effect of molnupiravir-induced mutations

High G-to-A branches made up a considerable percentage of branches involving more than ten substitutions in some countries (Fig. 5a), suggesting that molnupiravir drives a substantial proportion of large saltations. We found that high G-to-A branches had a different distribution of branch lengths from other types of branches. In typical SARS-CoV-2 evolution, the branch length distribution contains many more nodes with shorter branch lengths than with larger branch lengths; however, for nodes satisfying the high G-to-A criterion this decline was much less pronounced, with long-branch lengths still relatively common (Fig. 5b).

We also examined whether the mutations identified induced changes to amino acid sequence (non-synonymous mutations) or not (synonymous mutations). We found that for short branches in the tree, 65% (64.6–64.7%) of mutations in the spike (*S*) gene were non-synonymous. For long branches (ten or more mutations) that lacked the high G-to-A signature, the proportion of *S* mutations that

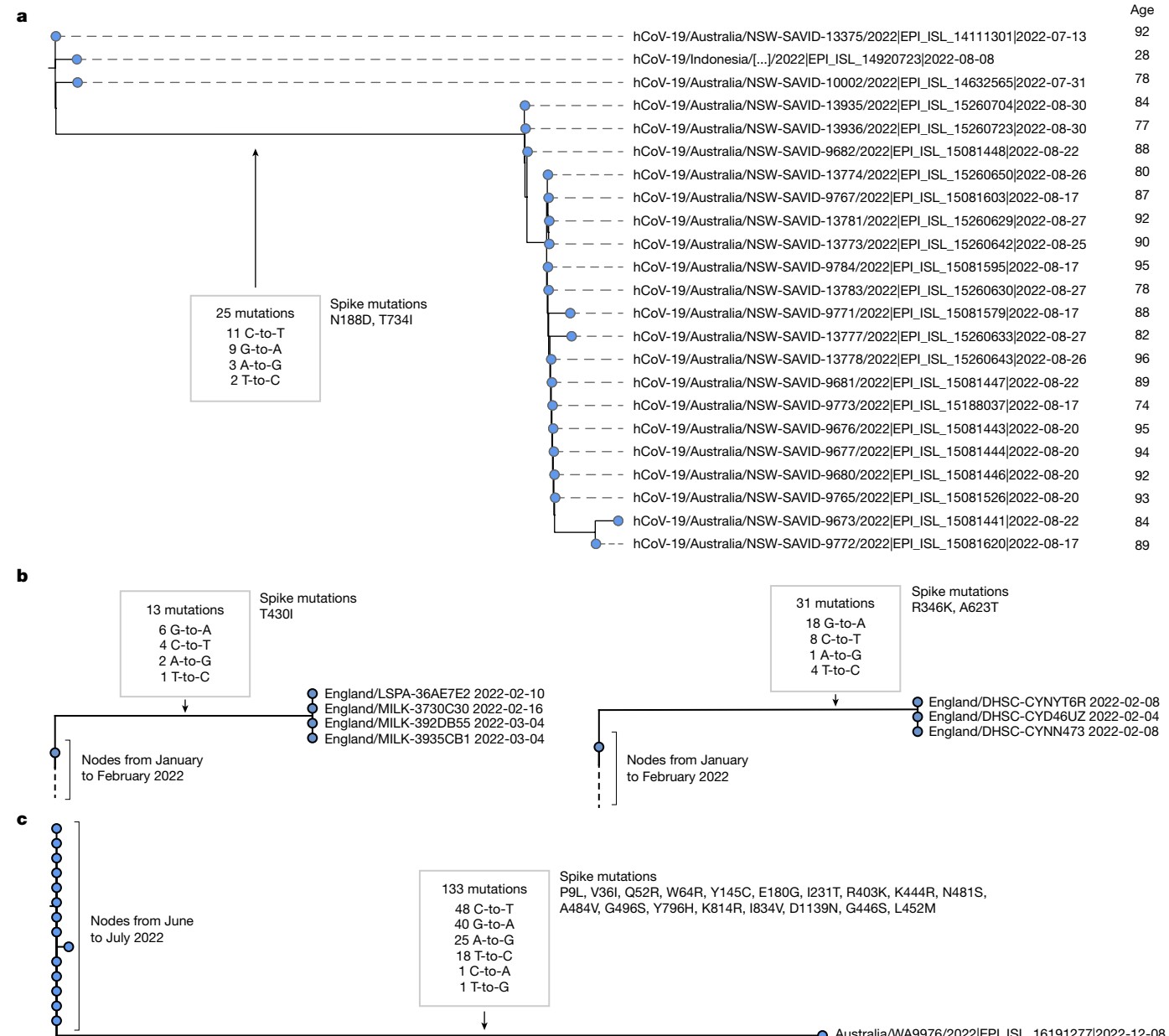

**Fig. 4 | High G-to-A branches can be associated with transmission clusters and, separately, can involve more than 100 mutations. a**, A cluster of 20 individuals emerging from a high G-to-A mutation event. This cluster involves a saltation of 25 mutations occurring within approximately one month, all of which are transition substitutions, with an elevated G-to-A rate. Sequences were annotated with age metadata suggestive of an outbreak in an aged care facility. Phylogenetic placement within the cluster is affected by missing coverage in some regions. **b**, Examples of further transmission clusters from the UK. Left, four sequences from the UK from February to March 2022 with 13 shared mutations with the high G-to-A signature. Right, a cluster of four

sequences from the UK from February 2022 with 31 shared mutations with the high G-to-A signature. **c**, A sequence from Australia with a high G-to-A signature and a total of 133 mutations relative to the closest outgroup sequence. Just 2 of the 133 mutations observed were transversions; transitions included many G-to-A events. (In the month after this sequence was deposited, two additional related or descendant sequences, EPI_ISL_16315710 and EPI_ISL_16639468, were deposited, which may represent continued sampling from the same patient as they involve a substantial subset of shared mutations, but not full concordance, which is suggestive of complex intrahost evolution).

were non-synonymous was higher at 77% (76.80–77.55%). This increase may reflect, in part, positive selection during intrahost evolution in individuals with chronic infections. In contrast, for long branches with a (molnupiravir-associated) high G-to-A signature, the proportion of mutations in the spike (S) protein that were non-synonymous was 63% (60.3–65.1%), similar to that of short branches, and substantially lower than that of other long branches ($P < 0.001$).

Despite this overall indication of purifying selection, which is consistent with the actions of a mutagenic drug, there was also evidence

for positive selection. Even in high G-to-A branches, there was a concentration of non-synonymous mutations in *S*, especially among the most recurrent mutations (Fig. 5c). Many of the recurrent *S* mutations, such as S:P9L, S:A701V, S:K147E, S:R493Q and S:G252S, were also mutations that arise in variants of concern or chronic infections, including S:E340K, which has been associated with sotrovimab resistance (Fig. 5d). There was good correspondence between the contexts in which these mutations occurred and the molnupiravir mutation spectrum. For example, the most common context for G-to-A class

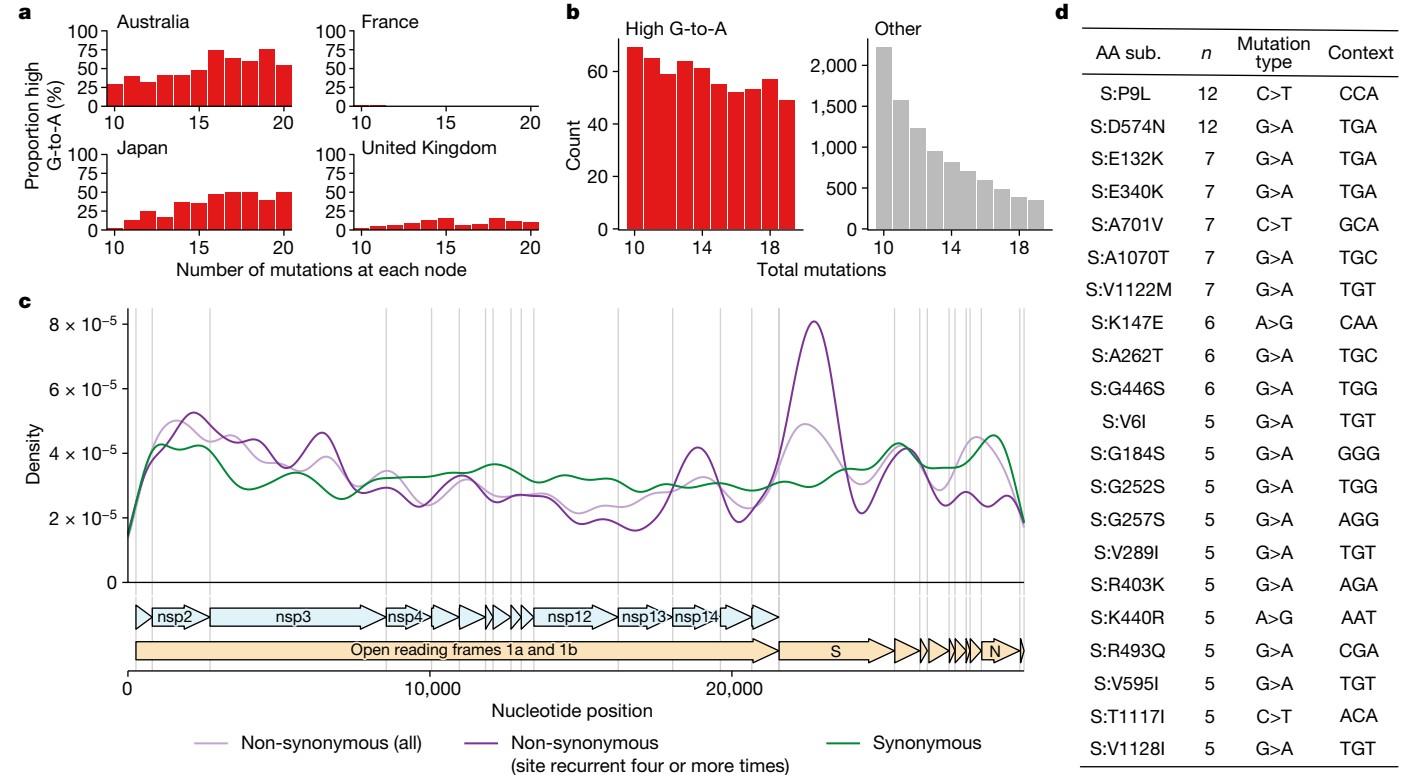

**Fig. 5 | High G-to-A branches make up a considerable proportion of long branches in affected countries and include evidence of selection.** **a**, Proportion of branches that are high G-to-A for a range of branch lengths in different countries. Data are from collection dates in 2022 and 2023 (submission dates up to June 2023). **b**, Branch length distributions for high G-to-A and other branches. **c**, Genomic distribution of mutations in high G-to-A nodes, partitioned into three classes: synonymous mutations; non-synonymous mutations; and non-synonymous mutations that occur four or more times. **d**, Table of the most recurrent mutations in *S* in high G-to-A branches. *n* = the number of high G-to-A branches exhibiting the mutation. Mutation type shows the parental and final nucleotide at the nucleotide position driving the mutation, while context shows the trinucleotide context for the mutated nucleotide, transcribed assuming a NC_045512.2 background.

mutations among those listed in Fig. 5d is TGT, which has a high enrichment in the molnupiravir spectrum and a low enrichment in the normal BA.1 spectrum (Extended Data Fig. 5c).

There was also a relative concentration of recurrent non-synonymous mutations in the exonuclease encoded by *nsp14*. This proofreading exonuclease functions to correct errors during genome replication, but typically has poor performance in recognizing mismatches involving molnupiravir[33]. Future work could examine whether there is a relationship between specific mutations in *nsp14* and tolerance to molnupiravir.

## Confirmation from treatment records

To better test a direct relationship between high G-to-A branches and the use of molnupiravir, linkage analysis was performed for high G-to-A branches sampled in England with treatment data in the Blueteq database[34]. This analysis found that 31% of clades descended from a high G-to-A branch involved an individual prescribed with molnupiravir (11 sequences in singleton clades, with sampling dates from day 7 to day 61 after the treatment start date). The overall rate of molnupiravir prescription in sequenced individuals in England from 2022 is 0.043%.

Not all branches analysed were linked to a person known to have been prescribed molnupiravir. In some cases, these could represent false positives in our analysis. In addition, the Blueteq database does not contain prescription data for people treated as part of clinical trials (which make up around a third of total molnupiravir prescriptions in the UK) or patients who fell outside interim clinical policy; it is also possible that in some cases we have not sampled the index case treated with molnupiravir, but instead an individual downstream of a treated patient in a transmission chain.

## Discussion

The observation that molnupiravir treatment has left a visible trace in global databases of consensus SARS-CoV-2 genomes, including onward transmission of molnupiravir-derived lineages, will be an important consideration for assessing the effects and evolutionary safety of this drug. Our results are consistent with recent observations in immuno-compromised individuals[35].

New variants of SARS-CoV-2 are generated through the acquisition of mutations that enhance properties, including immune evasion and intrinsic transmissibility[36,37]. The impact of molnupiravir treatment on the trajectory of variant generation and transmission is difficult to predict. A high proportion of induced mutations are likely to be deleterious or neutral, and it is important to consider a counterfactual to molnupiravir treatment that might involve higher viral load, potentially increasing the absolute number of diverse sequences[38,39]. However, molnupiravir increases per-sequence diversity in the surviving population, potentially with many mutations per genome, which might provide a broader substrate for selection to act on during intrahost evolution. Importantly, the divergence of the molnupiravir mutation spectrum from standard SARS-CoV-2 mutational dynamics might allow the virus to explore the fitness of distinctive parts of the possible genomic landscape to those that it is already widely exploring in the general population. Molnupiravir-induced mutations could also potentially allow infections to persist for longer by creating a more varied target for the immune system: this might contribute to the PANORAMIC trial's finding that while individuals treated with molnupiravir had a much lower viral load at day five, they had a slightly higher viral load than placebo-arm individuals at day 14 (ref. 2). Notably, in some countries a significant proportion of sequences

with the longest branch lengths are attributable to molnupiravir. However, at the time of writing, the largest clusters satisfying our criteria consist of approximately 20 sequenced individuals.

Considerations of the mechanism of action are important in the design and assessment of antiviral drugs. Molnupiravir's mode of action is often described using the term 'error catastrophe', the concept that there is an upper limit on the mutation rate of a virus beyond which it is unable to maintain self-identity[40]; however, this model has been criticized on its own terms[41] and is particularly problematic in the case of short-term antiviral treatment as it assumes an infinite time horizon. The 'lethal mutagenesis' model is much more useful in this context[42]. Not all nucleoside analogue drugs function primarily through mutagenesis. Many act through chain termination[43,44] and therefore would not be expected to cause the effects seen in this study for molnupiravir.

Our study illustrates the far-reaching potential of the extensive genomic dataset created by the community of researchers tracking SARS-CoV-2 evolution. The combination of all available global sequences increased the power of our analyses, while comparisons between countries with different treatment regimens were highly informative. Public health authorities should perform continued investigations into the effects of molnupiravir in viral sequences and the transmissibility of molnupiravir-derived lineages. These data will be useful for ongoing assessments of the risks and benefits of this treatment and may guide the future development of mutagenic agents as antivirals, particularly for viruses with high mutational tolerances, such as coronaviruses.

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

## Methods

### Processing of pre-existing genomic data from individuals treated with molnupiravir

We used three existing sources of genomic data in the calculations of the mutation classes (and later the contextual mutational spectra) associated with known use of molnupiravir and with typical SARS-CoV-2 evolution in the absence of molnupiravir. We analysed a dataset from Alteri et al.[17], which contained longitudinal data for both individuals treated with molnupiravir and untreated individuals. For this we downloaded FASTQ files from BioProject no. ERP142142 using fastq-dl[46]. We mapped these reads to the Hu-1 reference genome using minimap2 and then extracted the number of calls for each base at each position. We identified mutations compared to the day 0 sequence, counting variants where the site had 100 or more reads of which 5% or more were variants to the day 0 consensus. As a secondary dataset, we used data from the AGILE trial (Donovan-Banfield et al.[18], BioProject no. PRJNA854613). There was general agreement on the nature of molnupiravir mutations between Alteri et al.[17] and the AGILE trial, with the exception of a high G-to-T mutation rate seen only in the AGILE trial. Previous evidence on molnupiravir's mutation classes, as well as the fact that a high G-to-T rate was seen even in untreated individuals in the AGILE data, led us to conclude that this G-to-T signal in the AGILE data represented a technical artefact.

We used the BA.1 mutational spectrum previously calculated by Ruis et al.[19] as an exemplar of the mutation classes and spectrum under typical SARS-CoV-2 evolution in the relevant time period. To compare mutational burden according to mutation class between individuals treated with molnupiravir and untreated individuals, we scaled the Ruis et al.[19] dataset of typical BA.1 evolution to have the same number of total mutations as untreated individuals in the Alteri et al.[17] dataset, and then plotted these against individuals treated with molnupiravir from the Alteri et al.[17] dataset (Fig. 2a).

To identify which mutation classes were diagnostic of the use of molnupiravir, we first calculated what proportion of mutations came from each mutation class for both the Alteri et al.[17] molnupiravir dataset and the Ruis et al.[19] BA.1 dataset. We then calculated the ratio of these proportions between the molnupiravir and (naive) BA.1 datasets. To establish uncertainties in these ratios, we performed bootstrap resampling from each set of mutations (with 1,000 bootstrap repeats). These data are presented in Fig. 2b.

### Identification of high G-to-A sequences from UShER mutation-annotated tree

To identify sequences in global databases with a molnupiravir-like pattern of mutations, we analysed a regularly updated mutation-annotated tree built by the Ultrafast Sample placement on existing tRees (UShER) team[47] using almost all global data from INSDC and GISAID, a version of the McBroome et al. tree[22]. We extracted data using a script initially adapted from TaxoniumTools[48] and later modified to use the Big Tree Explorer (BTE)[49]. The script added metadata from sequencing databases to each node, then passed these metadata to parent nodes using simple heuristics: (1) a parent node was annotated with a year if all of its descendants were annotated with that year; (2) a parent node was annotated with a particular country if all of its descendants were annotated with that country; (3) a parent node was annotated with the mean age of its (age-annotated) descendants. Nodes with descendants spanning multiple years or countries were rare. We also calculated a more nuanced time estimate for nodes using Chronumental[50]. We used Taxonium[48], the UShER web interface[47], Nextstrain[51] and Nextclade[52] extensively in investigating individual branches of interest.

We defined 'high G-to-A branches' as those with at least ten mutations, of which more than 90% were transitions and more than 25% were specifically G-to-A mutations, with more than 20% C-to-T. Such a threshold yielded very high specificity, as judged by the ability to detect marked changes in the rate of a rare event (molnupiravir treatment) over time. We also created simulated measures of sensitivity and specificity using the distribution of mutation types from Ruis et al.[19] and Alteri et al.[17] We performed these calculations for different branch lengths ($n$) from ten to 20. In each case, we performed 10,000 draws of $n$ mutations from each of the naive and molnupiravir-associated mutational class distributions. We then assessed what proportion of these draws satisfied our criteria defined above. In the case of the molnupiravir-associated class distribution, this proportion represented the sensitivity. In the case of the typical BA.1 distribution, this proportion represented $1-$specificity. We obtained a sensitivity of 46% and a specificity of 98.9% for branch length 10, a sensitivity of 63% and a specificity of 98.6% for branch length 13, a sensitivity of 71% and a specificity of 98.6% for branch length 15 and a sensitivity of 64% and specificity of 99.8% for branch length 20.

To measure whether high G-to-A branches showed a statistically significant increase in mutation rate, we used Chronumental's branch length estimates in time and performed statistical testing with a two-sided $t$-test on nodes from 2022, looking only at nodes with at least ten mutations.

To test whether age metadata differed according to the presence of the high G-to-A signature, we took all US nodes from 2022 that were above the minimum branch length (10 or more) and divided them according to the presence or absence of the high G-to-A signature. We performed a two-sided $t$-test to test the significance of the effect seen. To verify that the effect was not substantially driven by the heuristic of taking the mean of descendant nodes, we repeated the analysis considering only branches with a single descendant and found highly similar results.

### Calculation of mutational spectra

To identify the preferred nucleotide contexts for molnupiravir-based mutagenesis, we calculated single-base substitution spectra. For the high G-to-A branches, we extracted mutation paths from the UShER mutation-annotated tree. The context of each mutation was identified using the Wuhan-Hu-1 genome (accession no. NC_045512.2), incorporating mutations acquired earlier in the path. Mutation counts were rescaled according to genomic content by dividing the number of mutations by the count of the starting triplet in the Wuhan-Hu-1 genome. MutTui (https://github.com/chrisruis/MutTui) was used to rescale and plot the mutational spectra.

To calculate a single-base substitution spectrum from the Alteri et al.[17] dataset, we used the mapped reads from BioProject no. PRJNA854613, again taking sites which had 100 or more reads of which 5% or more were distinct from the day 0 consensus. We rescaled mutation counts to mutational burdens by dividing each mutation count by the number of the starting triplet in the Wuhan-Hu-1 genome (accession no. NC_045512.2).

We performed a similar analysis for the Donovan-Banfield et al. data[18]. We used deep sequencing data from samples collected on day one (before treatment) and day five (after treatment) from 65 patients treated with placebo and 58 patients treated with molnupiravir. For each patient, we used the consensus sequence of the day one sample as the reference sequence and identified mutations as variants in the day five sample away from the patient reference sequence in at least 5% of reads at genome sites with at least 100-fold coverage. The surrounding nucleotide context of each mutation was identified from the patient reference sequence.

To ensure that any spectrum differences between placebo and molnupiravir treatments were not due to previously observed differences in spectrum between SARS-CoV-2 variants[19,20], we compared the distribution of variants between treatments (Extended Data Fig. 3). The distributions were highly similar.

We compared the contextual patterns within each transition mutation type, assessing the similarity of the values of the 16 possible trinucleotide context from the high G-to-A phylogenetic branches against those from the Alteri et al.[17] dataset and, separately, those from the Donovan-Banfield et al.[18] dataset and the Ruis et al.[19] control spectrum. For each dataset combination, cosine similarities were calculated for each transition mutational class. We performed the same correlational analysis within the long-branch data, comparing the G-to-A subset with the C-to-T subset, matching each G-to-A context to its reverse complement in the C-to-T dataset.

## Identifying highly divergent molnupiravir-derived sequences excluded from the mutation-annotated tree

Given that in the process of constructing the UShER mutation-annotated tree highly divergent sequences can be excluded, we decided to perform a secondary analysis to identify divergent sequences with a molnupiravir signature. We used Nextclade[52] for this task. We supplied a full dataset of full-length FASTA sequences; every sequence that could be aligned with Nextclade was included. Nextclade places each sequence onto a sparse reference phylogenetic tree. Its outputs include an 'unlabelled private mutations' column, which contains private mutations at a node with respect to the tree, excluding revertant mutations and mutations that are very common in other clades. We analysed this set of mutations for the presence of molnupiravir-like mutation class distributions.

We selected sequences that had 20 or more mutations of which 20% or more were G-to-A, 20% or more were C-to-T and 90% or more were transitions. Again, these were heavily enriched for dates after the roll-out of molnupiravir. We placed identified sequences onto a downsampled global tree using usher.bio and visualized this tree using Nextstrain[51].

To test whether 100 or more mutations in the sequence shown in Fig. 4c had contexts more compatible with the molnupiravir spectrum we identified in this study or with a typical BA.1 spectrum, we performed analysis using multinomial models. We aimed to ignore the signal from the mutation classes themselves (because these were used to select the sequence as interesting) and to consider only the extra information added by the contexts in which transition mutations occurred. For each transition class (G-to-A, C-to-T, A-to-G, T-to-C), we created two multinomial models of the trinucleotide context, one using the long-branch molnupiravir spectrum we defined in this study, and one using the BA.1 spectrum from Ruis et al.[19]. In each case, we multiplied by the number of times a trinucleotide context occurred in the genome to remove the previous normalization against this parameter. We assessed the likelihood of observing the counts of contexts in the sequence of interest under both models and calculated a Bayes factor for each (G-to-A: 35,017, C-to-T: 6,068, A-to-G: 53, T-to-C: 1.22). These combine to give a Bayes factor of $1.4 \times 10^{10}$.

## Analysis of synonymous and non-synonymous mutations

We examined the types of mutations that made up these branches. We used BTE to determine whether each mutation observed was synonymous or not. Mutations were tallied by this status, grouped according to whether the branch was short (fewer than ten mutations) or long (ten or more mutations), and whether it had a high G-to-A signature. We calculated the proportion of mutations that were non-synonymous in each case, calculated the binomial confidence intervals for these proportions and compared them using a two-sided test of equal proportions using the prop.test function in R.

We plotted the distribution of mutations across the genome for high G-to-A branches, split according to whether the mutations were synonymous or not, while also plotting the distribution specifically for the most recurrent non-synonymous mutations occurring in four or more high G-to-A branches. Kernel density estimates were made with a Gaussian kernel and a bandwidth of 500 bp.

## Processing and visualization of cluster trees

The bulk trees presented in the supplement were plotted from the UShER tree using ggtree[53].

For the cluster of the 20 individuals shown in Fig. 4a, we observed small imperfections in UShER's representation of the mutation-annotated tree within the cluster resulting from missing coverage at some positions. Therefore, we recalculated the tree that we display here. We took the 20 sequences in the cluster, and the three closest outgroup sequences, we aligned using Nextclade[52], calculated a tree using IQ-TREE[54] and reconstructed the mutation-annotated tree using TreeTime[55]. We visualized the tree using FigTree[56].

## Linkage analysis to treatment records

Forty-nine sequences with high G-to-A signatures from England, which fell into 35 clusters, were analysed by the UK Health Security Agency (UKHSA). Sequences were linked to the Blueteq treatment records[34] based on the NHS number. Linkage was established for all sequences. The analysis found that 11 of the 35 distinct clusters involved an individual prescribed molnupiravir, giving a cluster hit rate of 31%. Only sequences sampled after the treatment date were counted, with no upper time limit.

## Limitations

Our work has some limitations. Identifying a particular branch as possessing a molnupiravir-like signature is a probabilistic rather than absolute judgement: where molnupiravir creates just a handful of mutations (which trial data suggest is often the case), branch lengths will be too small to assign the cause of the mutations with confidence. Therefore, we limited our analyses to long branches. This approach may also fail to detect branches that feature a substantial number of molnupiravir-induced mutations alongside a considerable number of mutations from other causes (which might occur in chronic infections). Our approach to identifying molnupiravir-associated sequences used simple thresholding on the proportion of mutations on a branch with different classes of mutation. The simplicity of this approach, which does not make detection probability a function of branch length, enabled us to perform analyses such as looking at the distribution of branch length in different conditions; however, future analyses that increase sensitivity with more nuanced statistical approaches (with which we experimented, finding the simple method preferable in this first case for the flexibility it offered), as well as considering the contextual mutation spectrum itself as a signal for detection, will both be valuable in future work.

We discovered drastically different rates of molnupiravir-associated sequences according to country and suggest that this reflects in part whether, and how, molnupiravir is used in different geographical regions; however, there will also be contributions from the rate at which genomes are sequenced in settings where molnupiravir is used. For example, if molnupiravir is used primarily in aged care facilities and viruses in these facilities are significantly more likely to be sequenced than those in the general community, this will elevate the ascertainment rate of such sequences. Furthermore, it is probable that some included sequences were specifically analysed as part of specific studies because the samples demonstrated continued test positivity after molnupiravir treatment. Such effects are likely to differ based on sequencing priorities in different locations. We identified sequence clusters descending from high G-to-A nodes. In a number of cases, detailed and distinctive metadata showed that a particular cluster was made up of sequences from different patients, suggesting transmission of molnupiravir-induced mutations; however, in the absence of such data, clusters are also compatible with representing multiple samples taken from a single individual.

Our analysis looked at consensus sequences, which means that for a mutation to be detected it must reach a high proportion of the

population in the host. Analyses that look at deep sequencing data and mixed base-calls in consensus sequences, will be valuable.

## Reporting summary

Further information on research design is available in the Nature Portfolio Reporting Summary linked to this article.

## Data availability

No new primary data were generated for this study. We used data from consensus sequences available through GISAID and the INSDC[23,24], from the AGILE clinical trial[18], where genomic data were obtained from BioProject no. PRJNA854613 at the Sequence Read Archive, and from Alteri et al.[17] from BioProject no. ERP142142. The AGILE investigators were not involved in the analysis and preparation of this manuscript. Linkage analysis was performed within the UKHSA. Section 251 of the National Health Service Act 2006 permits UKHSA use of patient-level data for specific projects. The findings of this study are based on metadata associated with 15,572,413 sequences available on GISAID up to June 2023 and accessible at https://doi.org/10.55876/gis8.230110wz, https://doi.org/10.55876/gis8.230110db and https://doi.org/10.55876/gis8.230622mw (see also the Supplementary Information). The findings of this study are also based on 7,104,124 sequences from the INSDC (authors, metadata and sequences are available at https://www.ncbi.nlm.nih.gov/labs/virus/vssi/#/virus?SeqType_s=Nucleotide&VirusLineage_ss=SARS-CoV-2,%5C%20taxid:2697049&CreateDate_dt=2010-01-01T00:00:00.00Z%20TO%202023-05-31T23:59:59.00Z). We analysed these data in the form of a mutation-annotated tree, which was a version of the one developed by McBroome et al.[22]. featuring all available genomic data from the INSDC and GISAID. Data present in both databases were deduplicated during the construction of the mutation-annotated tree on the basis of sequence, name and metadata. We used a version standardized to GISAID sequence names and accessions for sequences present in both databases. A version of our analysis using only the INSDC subset of the tree, using INSDC naming conventions, is available at https://github.com/theosanderson/molnupiravir/tree/main/open_data_version. Processed open data are available at https://zenodo.org/record/8252388.

## Code availability

Our GitHub repository is located at https://github.com/theosanderson/molnupiravir. It is archived on Zenodo at https://doi.org/10.5281/zenodo.8101003.

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

**Acknowledgements** We thank all data contributors, that is, the authors and their originating laboratories responsible for obtaining the specimens, and their submitting laboratories for generating the genetic sequence and metadata and sharing via the GISAID initiative, on which this research is based. We also thank everyone who contributed to the generation of the genomes deposited in the INSDC databases, on which this research is also based. We thank A. Hinrichs and colleagues for access to an UShER mutation-annotated tree built with all available genomic data. We thank NHS England for providing the Blueteq data on treatment records. We thank the UKHSA COVID-19 Therapeutics Programme Team past and present, in particular J. Charlesworth, A. Lackenby, A. Demirjian, M. Chand and C. Brown. We thank J. Bloom, M. Lin, R. Neher, K. Harris and F. Débarre for useful discussions. T.S. was supported by the Wellcome Trust (no. 210918/Z/18/Z) and the Francis Crick Institute, which receives its core funding from Cancer Research UK (no. FC001043), the UK Medical Research Council (MRC) (no. FC001043) and the Wellcome Trust (no. FC001043). This research was funded in whole, or in part, by the Wellcome Trust (nos. 210918/Z/18/Z, FC001043). For the purpose of open access, the authors have applied a CC-BY public copyright licence to any author-accepted manuscript resulting from this article. I.D.-B. is supported by PhD funding from the National Institute for Health and Care Research (NIHR) Health Protection Research Unit in Emerging and Zoonotic Infections at the University of Liverpool in partnership with Public Health England (now UKHSA), in collaboration with the Liverpool School of Tropical Medicine and the University of Oxford (award no. 200907). The views expressed are those of the authors and not necessarily those of the Department of Health and Social Care or NIHR. Neither the funders nor the trial sponsor were involved in study design, data collection, analysis, interpretation or preparation of the manuscript. G2P-UK National Virology Consortium, which is funded by the MRC (no. MR/W005611/1). C.R. was supported by a Fondation Botnar Research Award (programme grant no. 6063), the UK Cystic Fibrosis Trust (Innovation Hub Award 001) and funding from the Oxford Martin School.

**Author contributions** R.H. identified the initial branches and their likely connection to molnupiravir. T.S. performed the analyses of the mutation-annotated tree and global metadata. C.R. led the analyses of the mutational spectra. I.D.-B. created the bioinformatic pipelines for the AGILE trial data. T.P.P. functionally curated the mutations identified in the long branches. H.H. and A.L. performed the linkage analyses. All authors participated in manuscript writing.

**Funding** Open Access funding provided by The Francis Crick Institute.

**Competing interests** The authors declare no competing interests.

**Additional information**
**Correspondence and requests for materials** should be addressed to Theo Sanderson or Christopher Ruis.

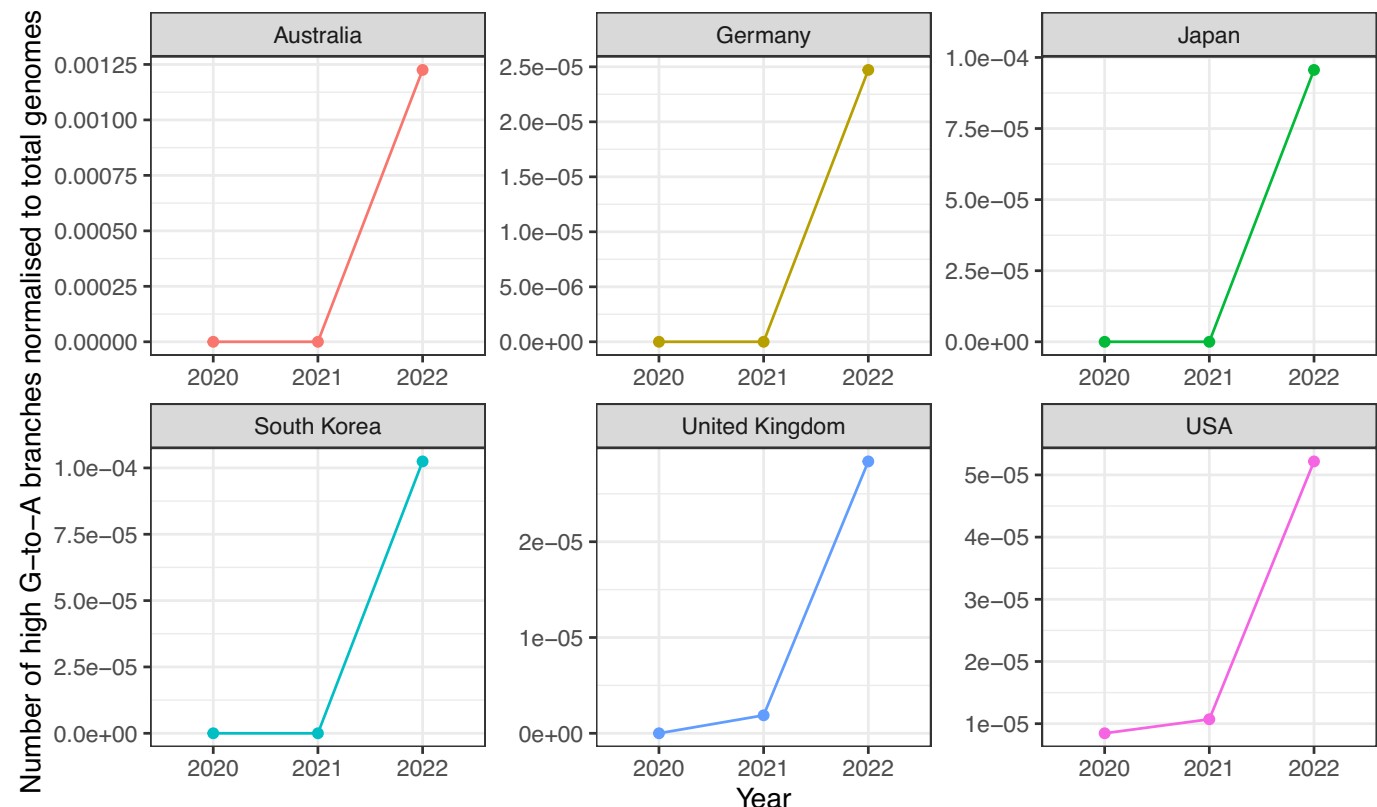

**Extended Data Fig. 1 | Timeline of number of high G-to-A branches, normalised for sequencing volumes, in 6 countries.** The y-axis represents number of high G-to-A branches, divided by total sequencing volume for the year. This analysis demonstrates that the effects seen in raw numbers in Fig. 2d cannot be explained by changes in sequencing volume.

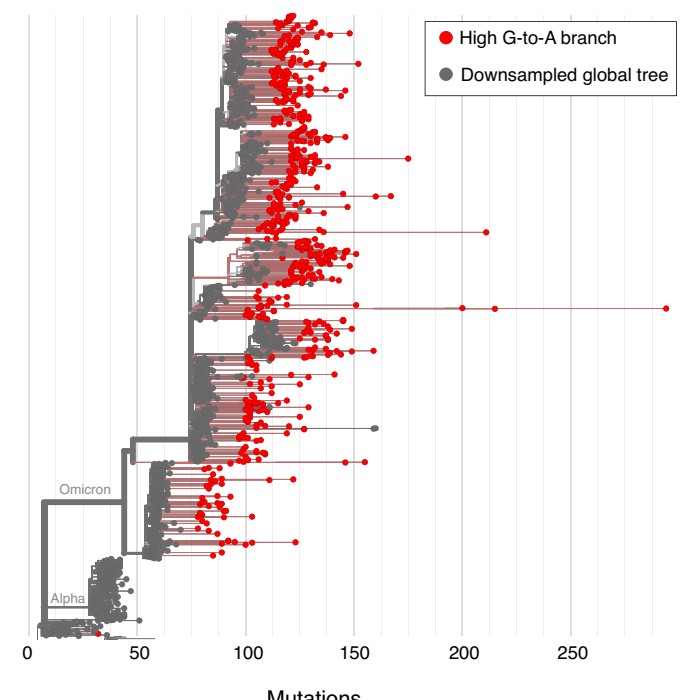

**Extended Data Fig. 2 | High G-to-A sequences with more than 20 private mutations identified from a Nextclade alignment of all available SARS-CoV-2 sequences.** Nextclade was used to align sequences and identify private mutations. High G-to-A branches were identified on the basis of unlabelled private mutations. Usher.bio was then used to create a tree with high G-to-A branches highlighted on a downsampled global tree, with visualisation performed with Nextstrain.

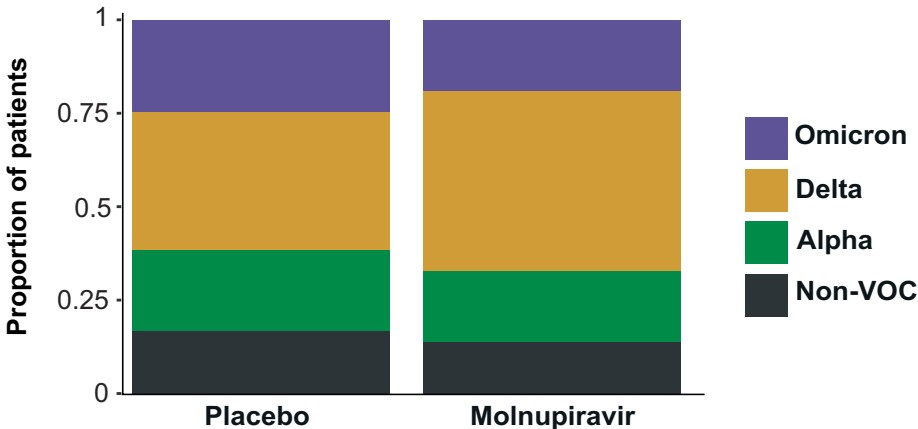

**Extended Data Fig. 3 | Distribution of major SARS-CoV-2 variants between placebo and molnupiravir treatments in the AGILE trial dataset.** The proportion of patients infected with each variant is shown. The proportions are similar suggesting that differences between placebo and molnupiravir spectra will not be influenced by previously observed spectrum differences between variants (Ruis et al., Bloom et al.). VOC = variant of concern.

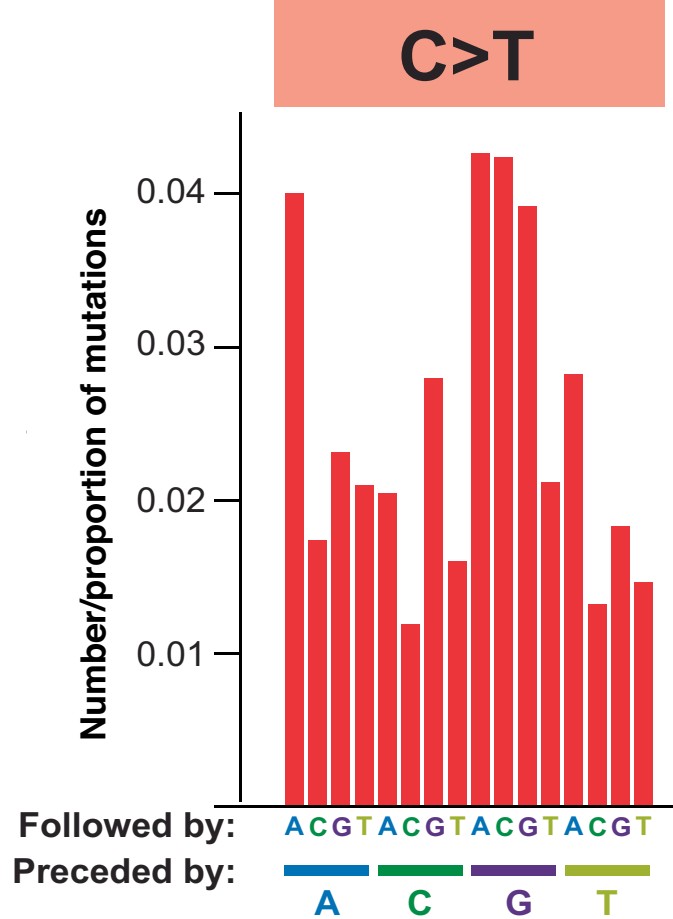

**Extended Data Fig. 4 | Context locations within the mutational spectrum.**
The RNA mutational spectrum contains 12 mutation types, for example C-to-T, shown here. The spectrum also captures the nucleotides surrounding each mutation. There are four potential upstream nucleotides and four potential downstream nucleotides. This figure shows the location of each of the 16 contexts within an example mutation type. For example, the leftmost bar represents C-to-T mutations in the ACA context while the second leftmost bar represents C-to-T mutations in the ACC context. The spectrum represented is from AGILE trial data on monlupiravir.

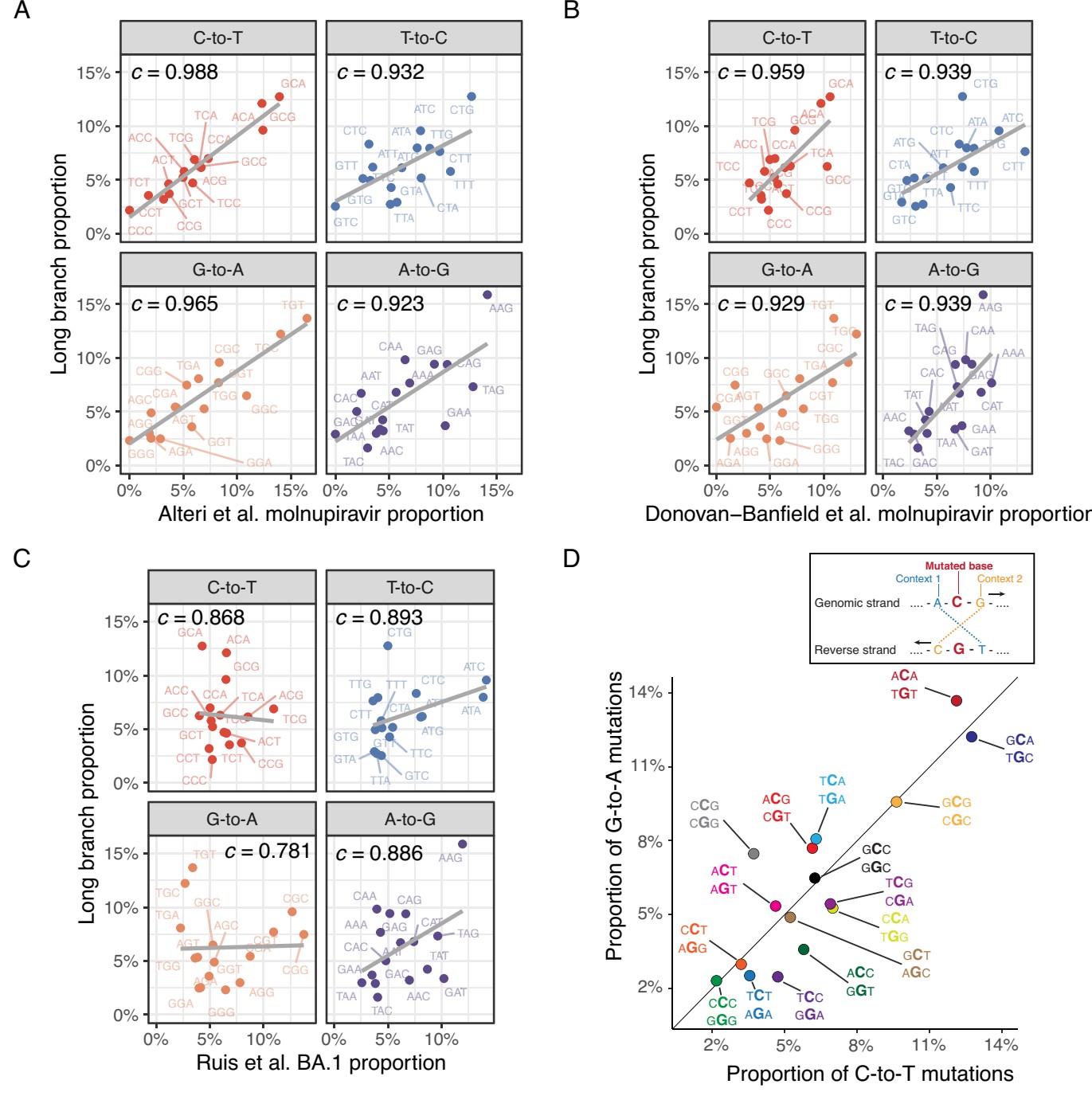

**Extended Data Fig. 5 | Mutation spectrum analysis supports a molnupiravir origin for high G-to-A nodes.** (**A**) Strong correlation for contexts in all transition mutation classes between Alteri et al. molnupiravir-treated patients and high G-to-A long branches. (**B**) Similar analysis, with clear correlation between Donovan-Banfield et al. dataset of molnupiravir treated individuals to long high G-to-A branches. (**C**) Little correlation seen between contexts in typical SARS-CoV-2 evolution (Ruis et al.) and high G-to-A branches. (**D**) In data from long branches, context proportions for G-to-A mutations correlate with context proportions for C-to-T mutations, indicating a common mutational process. Points are labelled with G-to-A context, then C-to-T context.

**Extended Data Table 1 | Number of high G-to-A branches from 2022 compared against the total number of genomes from 2022 by country**

| Country | branches in 2022 | in 2022 |
|---|---|---|
| Australia | 149 | 121,602 |
| USA | 106 | 2,031,795 |
| United Kingdom | 35 | 1,232,969 |
| Japan | 35 | 366,060 |
| Russia | 17 | 41,416 |
| Germany | 13 | 525,967 |
| Italy | 13 | 70,555 |
| Israel | 12 | 108,770 |
| Slovakia | 11 | 26,461 |
| Denmark | 10 | 332,006 |
| Thailand | 9 | 24,338 |
| Austria | 8 | 46,962 |
| Spain | 7 | 95,635 |
| New Zealand | 7 | 25,170 |
| South Korea | 6 | 58,567 |
| Turkey | 6 | 20,754 |
| France | 4 | 328,527 |
| Czech Republic | 4 | 32,124 |
| Ireland | 3 | 48,704 |
| Indonesia | 3 | 34,499 |
| Luxembourg | 3 | 30,715 |
| India | 2 | 121,841 |
| Belgium | 2 | 84,600 |
| Philippines | 2 | 12,830 |
| Canada | 1 | 217,040 |
| Sweden | 1 | 88,418 |
| Poland | 1 | 44,014 |
| Mexico | 1 | 35,857 |
| Slovenia | 1 | 31,221 |
| Norway | 1 | 30,796 |
| South Africa | 1 | 15,502 |
| Latvia | 1 | 14,039 |
| Hong Kong | 1 | 10,969 |
| Brazil | 0 | 98,346 |
| Netherlands | 0 | 64,614 |
| Switzerland | 0 | 49,062 |
| Peru | 0 | 30,772 |
| Malaysia | 0 | 27,113 |
| Croatia | 0 | 22,786 |
| Chile | 0 | 20,229 |
| Portugal | 0 | 19,483 |
| Finland | 0 | 18,518 |
| Singapore | 0 | 17,353 |
| Greece | 0 | 14,293 |
| Colombia | 0 | 12,302 |
| China | 0 | 11,425 |
| Lithuania | 0 | 10,059 |

Only countries with more than 10,000 total genomes in 2022 are included.

# Reporting Summary

## Statistics

For all statistical analyses, confirm that the following items are present in the figure legend, table legend, main text, or Methods section.

| n/a | Confirmed | |
|---|---|---|
| ☐ | ☒ | The exact sample size (*n*) for each experimental group/condition, given as a discrete number and unit of measurement |
| ☒ | ☐ | A statement on whether measurements were taken from distinct samples or whether the same sample was measured repeatedly |
| ☐ | ☒ | The statistical test(s) used AND whether they are one- or two-sided<br>*Only common tests should be described solely by name; describe more complex techniques in the Methods section.* |
| ☐ | ☒ | A description of all covariates tested |
| ☐ | ☒ | A description of any assumptions or corrections, such as tests of normality and adjustment for multiple comparisons |
| ☐ | ☒ | A full description of the statistical parameters including central tendency (e.g. means) or other basic estimates (e.g. regression coefficient) AND variation (e.g. standard deviation) or associated estimates of uncertainty (e.g. confidence intervals) |
| ☐ | ☒ | For null hypothesis testing, the test statistic (e.g. *F*, *t*, *r*) with confidence intervals, effect sizes, degrees of freedom and *P* value noted<br>*Give P values as exact values whenever suitable.* |
| ☒ | ☐ | For Bayesian analysis, information on the choice of priors and Markov chain Monte Carlo settings |
| ☒ | ☐ | For hierarchical and complex designs, identification of the appropriate level for tests and full reporting of outcomes |
| ☒ | ☐ | Estimates of effect sizes (e.g. Cohen's *d*, Pearson's *r*), indicating how they were calculated |

*Our web collection on statistics for biologists contains articles on many of the points above.*

## Software and code

Policy information about availability of computer code

| Data collection | Code was not used to collect data in this study. |
|---|---|
| Data analysis | We have made a version of our analysis that uses a fully open dataset (therefore covering with fewer sequences) and open source code available at https://github.com/theosanderson/molnupiravir . This is available at Zenodo at https://zenodo.org/record/8309773 .<br><br>Chronumental v0.0.60 was used<br>Nextclade v2.12.0 was used<br>BTE v0.9.0 was used<br>MutTui v2.0.2 was used<br><br>Full R package versions available at  https://github.com/theosanderson/molnupiravir/blob/main/archive/package_versions.txt |

For manuscripts utilizing custom algorithms or software that are central to the research but not yet described in published literature, software must be made available to editors and reviewers. We strongly encourage code deposition in a community repository (e.g. GitHub). See the Nature Portfolio guidelines for submitting code & software for further information.

## Data

No new primary data was generated for this study. We used data from consensus sequences available through GISAID and the INSDC24,25, from the AGILE clinical trial 20, where genomic data were obtained from BioProject PRJNA854613 at the SRA, and from Alteri et al 18 from BioProject ERP142142. The AGILE investigators were not involved in the analysis and preparation of this manuscript. Linkage analysis was performed within UKHSA. Section 251 of the National Health Service Act 2006 permits UKHSA use of patient-level data for specific projects. The findings of this study are based on metadata associated with 15,572,413 sequences available on GISAID up to June 2023, and accessible at 10.55876/gis8.230110wz, and 10.55876/gis8.230110db, 10.55876/gis8.230622mw (see also, Supplemental Tables). The findings of this study are also based on 7,104,124 sequences from INSDC.

We analysed these data in the form of a mutation-annotated tree, which is a version of McBroome et al. Data present in both databases are deduplicated during the construction of the mutation-annotated tree on the basis of sequence, name, and metadata. We standardised to GISAID sequence names and accessions for sequences present in both databases.

A version of our analysis using only the INSDC subset of the tree, with INSDC naming conventions, is available at https://github.com/theosanderson/molnupiravir/tree/main/open_data_version. Processed open data is available at https://zenodo.org/record/8252388 .

## Human research participants

| Reporting on sex and gender | N/A |
|---|---|
| Population characteristics | N/A |
| Recruitment | N/A |
| Ethics oversight | N/A |

Note that full information on the approval of the study protocol must also be provided in the manuscript.

# Field-specific reporting

Please select the one below that is the best fit for your research. If you are not sure, read the appropriate sections before making your selection.

☒ Life sciences    ☐ Behavioural & social sciences    ☐ Ecological, evolutionary & environmental sciences

For a reference copy of the document with all sections, see nature.com/documents/nr-reporting-summary-flat.pdf

# Life sciences study design

All studies must disclose on these points even when the disclosure is negative.

| Sample size | We used all available data |
|---|---|
| Data exclusions | No data were excluded |
| Replication | This study used all available data. We replicate results in the sense that similar effects are seen in multiple different countries that use molnupiravir. We did not conduct experiments but analysed data that was available in databases. |
| Randomization | This study did not have experimental groups. We analysed all available data. There was no opportunity for randomization. |
| Blinding | No blinding was performed. Blinding was not necessary given the large number of sequences being analysed and the limited degrees of freedom available. |

# Reporting for specific materials, systems and methods

We require information from authors about some types of materials, experimental systems and methods used in many studies. Here, indicate whether each material, system or method listed is relevant to your study. If you are not sure if a list item applies to your research, read the appropriate section before selecting a response.

## Materials & experimental systems

| n/a | Involved in the study |
|---|---|
| ☒ | ☐ Antibodies |
| ☒ | ☐ Eukaryotic cell lines |
| ☒ | ☐ Palaeontology and archaeology |
| ☒ | ☐ Animals and other organisms |
| ☒ | ☐ Clinical data |
| ☒ | ☐ Dual use research of concern |

## Methods

| n/a | Involved in the study |
|---|---|
| ☒ | ☐ ChIP-seq |
| ☒ | ☐ Flow cytometry |
| ☒ | ☐ MRI-based neuroimaging |

