## [Peer Review File · Nature]

Manuscript Title: A molnupiravir-associated mutational signature in global SARS-CoV-2 genomes

Reviewer Comments & Author Rebuttals

Reviewer Reports on the Initial Version:

Referees' comments:

Referee #1 (Remarks to the Author):

The article by Sanderson et al entitled "Identification of a molnupiravir-associated mutational signature in SARS-CoV-2 sequencing databases" describes an emerging pattern of hypermutated, G>A dominated branches emerging in international SARS-CoV-2 sequencing data after introduction of molnupiravir treatment. Molnupiravir is a nucleotide analogue, which is incorporated during RNA transcription; its mechanism of action is its mutational effect which damages the resulting virus.

This is a timely and well-conducted analysis highlighting a worrying pattern of hypermutated viruses caused by molnupiravir entering circulation in the broader population. This bears the risk of accelerating the evolution of new viral lineages. It is only a mildly comforting signs that transmission of detected variants was only short-lived.

The evidence presented in the paper is a smoking gun.

I have only minor comments related to the analysis.

1. I wondered whether the epidemiological analysis related to the usage patterns of molnupiravir and emergence of hypermutated branches could be further strengthened. It is striking to see that most of such hypermutated branches emerged in countries with high use (Figure 3a). I wondered whether one could conduct a calculation that reveals how often a course of molnupiravir treatment caused a new hypermutated variant. The authors noted that there were 380,000 prescriptions recorded in Australia leading to 100 hypermutated events. Based on an guesstimated attack rate of 50% in 2022, there were approximately 13 million infections in Australia of which 100,000 (~1%) were sequenced. Based on this level of sequencing coverage one would expect a total of around $100 / 0.01 = 10,000$ true hypermutation events, or around 1/38 courses of treatment. (My back of envelope calculation may be wrong, but the trend seems worrying)

2. It may be useful to more explicitly contrast the hypermutation pattern (which is uniform across the genome and has $dN/dS = 1$) from that of hyperselected variants of concern, as in Omicron BA.1, which has a $dN/dS \gg 1$ with most coding changes in the S gene. That would strengthen the argument that the molnupiravir pattern is very different (and thus clearly discernible) from that of other means of natural selection.

3. The exposition of the manuscript could be streamlined a little, but I commend the authors for conducting their analysis in a very short amount of time. The manuscript meanders somewhat between molnupiravir's biochemical mechanism of actions, mutation patterns seen in data, epidemiology back to mutation patterns in the clinical trial.

Referee #2 (Remarks to the Author):

This is a really nice paper on an important topic. It has potentially substantial public health impact, and so I recommend publication as rapidly as possible (though I note and commend the authors for preprinting these findings alongside peer review).

Statistical tests are appropriate.

Detailed comments are below.

Major comments:

1. At present (presumably because this is the order of the work done by the authors) we get molnupiravir mutational theory, then the population data, then the direct examination of the drug's mutations from the AGILE data. I think the paper might flow better if the AGILE data is brought to the front, right after the intro on the theory, and then the population data are presented. Then we get: theoretical expectation of molnupiravir treatment, confirmation of it in trial data, and then search for that signature in the real world.

2. Following on from the point above, the authors used heuristics (20 subs to find long branches, then 10 subs with 25% G->A, 20%C->T, low transversions), again, I'm guessing because that's how they explored the GISAID data initially. I'm sure this gets at the key signal, but is there a more sophisticated way to find the molnupiravir-like branches, esp. knowing the trial data signature? e.g. there's a lot of 9 sub branches (Fig 4 suggests the length decays exponentially), and even a simple probabilistic model using the signatures might yield a lot more data to analyse.

3. I think the authors need some precision about what group of "long branches" they are talking about at different points. For example on page 5 the authors examine spectra on "these branches". Is the antecedent the previous paragraph (v. long branches found specifically in Australia), or the aforementioned heuristic (10 subs with 25% G->A, 20%C->T, low transversions), which by definition will have G->A and C->T mutations, or something else? Similarly for Fig S7, which set of long branches is this?

4. Figure 3C needs some more explication for me. There does seem to be a relationship between number of sequences and number of long branches. Saying that the latter "could not be explained" by the former and then pointing to that graph needs more information. (Since it is log-log, it can be hard to realise how big an outlier, e.g. Australia and France, are). Actually without the logs, does it make the point more clearly?

Minor comments:

- Abstract: "approach"
- Abstract: "largely corresponds" is a bit weaselly. Be more precise?
- Figure S1 is nice, maybe pull it as a panel along Fig 1 & 2 into one place in main text?
- Figure 3A: "G-to-A ratio" Ratio of what? I think it is G-to-A to everything from the legend, but then I think "proportion" would be easier to understand than "ratio"? I think in other contexts "transition:transversion" ratio is common, but here again I think this is transitions:total (i.e. a proportion)
- Figure 3B: does this look any different if considered as a proportion, rather than absolute number? There are way more 2022 and 2021 sequences than 2020, as countries got their pipelines up and running.
- Figure S2: why does the blue line in the UK seem to have a different slope? (and/or why does the USA have no slope?)
- Figure 5A: The text says "closely related outgroups" of the long branch come from July 2022. Are there no dates for this branch itself?
- Figure 7A: Is it surprising (or not) that there are similar levels of C>T and G>A? Does the virus spend more or less time in the relevant phase of the replication cycle?

- Figure 7B: It took me a couple of tries at the legend to understand this plot. At first I thought it was proportions of each mutation type among mutations, with each dot being I'm not sure what. Maybe be explicit in legend: "each dot is a trinucleotide context". Could even call out one dot in the plot "TGT or ACA", or similar?
- Figure 8C: there are more of nearly all kinds of mutations in the treatment arm. Any idea why? Direct molnupiravir effect not in the canonical places from Fig 2? Indirect effect of initial mutations on weaker error control? Something else?
- Figure S4 is not referenced until the methods, and Figure S5 is not referenced anywhere.
- Methods: the metadata adding heuristics seem like they have some problematic edge cases (e.g. at calendar breaks). I doubt it makes any difference, but the authors might want to comment.
- I found the layout of the PDF absolutely delightful to review!

Referee #3 (Remarks to the Author):

Sanderson et al present a retrospective computational phylogenetic analysis of over 13 million SARS-CoV-2 genomes from GISAID. They identify a set of unusually mutated sequences, bearing a particular signature enriched for G>A and C>T mutations, and propose that these sets may have been the result of intra-host evolution subject to (ineffective) applications of a mutagenising agent molnupiravir. There are three lines of evidence offered to justify this connection

- 1). That these mutational signatures are consistent with what would be expected from the molecular basis of action of molnupiravir.
- 2). That these mutations were contemporaneous with the approval and large-scale administration of molnupiravir in some (but not all countries), and that the set of sequences bearing these mutations was enriched for older patients (where such data are available).
- 3). That these mutations are largely (but not 100%) consistent with a study that performed direct sequencing of subjects enrolled in an RCT examining the efficacy of molnupiravir.

This is an intriguing study which is innovative and insightful in the following ways

- 1). The use of a large scale genomic database and "big-data" phylogenetic tools to uncover previously under-appreciated but biologically important patterns.
- 2). Indirect evidence of unexpected and potentially highly undesirable cases of drug-escape and accelerated evolution of RNA viruses, including evidence of onward transmission.

The second point above is of broad interest because it showcases how unanticipated effects of treatment with "unique" therapeutic agents may lead to the unexpected diversity of the targeted pathogen.

There are, however, several issues with the study.

- 1). The conclusions of the study rest on two big assumptions.
 - (a) Since we have limited knowledge of what "in the field" molnupiravir-induced evolution would look like, we must accept that G>A and C>T "hypermutations" are in fact attributable to molnupiravir-driven mutagenesis. I think the authors could have done considerably more here to further support this claim
 - (b) An even more severe limitations is that there are no meta-data on treatment histories in GISAID, hence we don't know which of the individuals with hyper-mutated sequences have been exposed to molnupiravir
 - (c) Because GISAID does not readily make underlying NGS data available, quality control and

validation of such hyper-mutated genomes (which would be requisite to rule out batch artifacts, and other technical issues) is impractical. Thus one is left to wonder (especially with cases like shown in Fig 6C with > 100 mutations) if these sequences are trustworthy.

(d) I am quite concerned that the authors did not really address the G>T mutation which IS enriched in the trial data, but NOT enriched in their GISAID clusters.

These issues makes causality impossible to establish, and muddies even associations, because we don't know the true "labels" for the sequences. While some of the indirect evidence (timing, association of molnupiravir availability with hypermutations) is suggestive, it is not sufficient to draw any strong conclusions.

2). Even if the discovered hyper-mutated clusters are attributable to molnupiravir, the manuscript does nothing at all to attempt to establish the significance of this. Are these mutations in any way likely to create altered viral phenotypes with a potential selective advantage (including resistance to molnupiravir)? Seeing how effectively all of the detected events are dead-ends, this is not very likely.

3). The manuscript needs to improve its statistical rigour. This takes many forms

(a). Lack of any formal statistical testing for many visual associations (e.g. Figure 3 C and D)

(b). Seemingly arbitrary definitions of mutational signatures in terms of G/A C/T ratios etc

(c). Lack of any kind of explicit null. It would be highly informative to estimate, using for example some available models of mutational preferences/biases for SARS-CoV-2, how likely are "natural" processes to generate particular features mutational signatures. As it is, there is no quantitative basis to evaluate "deviations from expected values"

(d). Missing data is a major issue here that is not addressed; for example Figure 3D comparison implies that age data are missing entirely at random.

Specific suggestions on further analyses.

1. Mutational signatures (i.e. counts of nucleotide mutations) form a very crude measure.

Evolution due to molnupiravir might leave other, detectable signatures

(a). What is the distribution of synonymous and non-synonymous mutations? Molnupiravir might induce a "neutral-like" evolutionary pattern. Alternatively, in order to survive, the putative molnupiravir sequences you see could be enriched for synonymous mutations. If you look at the mutations, what do they look like? Do you see any in RBD that have any significance (e.g. nAb or immune escape) and that have not been seen in circulating isolates? Do you see unusual indels in these sequences (not expected to be caused by molnupiravir)

(b). What do the spatial distribution of mutations look like? If randomly generated, they should NOT look like the mutations driving variant evolution, i.e. be different from others

(c). What about the other long branches (Figure 4)? They are not discussed at all.

2. Sensitivity analysis with respect to how these mutational signatures are defined.

3. More careful attention to interpretation and discussion. For example, assuming that a molnupiravir lineage does survive and even spread, what do you expect to happen? Figure 5A for example shows some odd patterns which may look like reversions (LATER sequences are closer to the MRCA than EARLIER sequences). What other processes could have given rise to these patterns? Can you better correlate the intensity of treatment with the # of clusters? Do different SC2 lineages experience similar/different levels of molnupiravir-induced mutagenesis?

Author Rebuttals to Initial Comments:

Referee #1

The article by Sanderson et al entitled “Identification of a molnupiravir-associated mutational signature in SARS-CoV-2 sequencing databases” describes an emerging pattern of hypermutated, G>A dominated branches emerging in international SARS-CoV-2 sequencing data after introduction of molnupiravir treatment. Molnupiravir is a nucleotide analogue, which is incorporated during RNA transcription; its mechanism of action is its mutational effect which damages the resulting virus.

This is a timely and well-conducted analysis highlighting a worrying pattern of hypermutated viruses caused by molnupiravir entering circulation in the broader population. This bears the risk of accelerating the evolution of new viral lineages. It is only a mildly comforting sign that transmission of detected variants was only short-lived.

The evidence presented in the paper is a smoking gun.

I have only minor comments related to the analysis.

1. I wondered whether the epidemiological analysis related to the usage patterns of molnupiravir and emergence of hypermutated branches could be further strengthened. It is striking to see that most of such hypermutated branches emerged in countries with high use (Figure 3a). I wondered whether one could conduct a calculation that reveals how often a course of molnupiravir treatment caused a new hypermutated variant. The authors noted that there were 380,000 prescriptions recorded in Australia leading to 100 hypermutated events. Based on an guesstimated attack rate of 50% in 2022 ,there were approximately 13 million infections in Australia of which 100,000 (1%) were sequenced. Based on this level of sequencing coverage one would expect a total of around $100 / 0.01 = 10,000$ true hypermutation events, or around 1/38 courses of treatment. (My back of envelope calculation may be wrong, but the trend seems worrying)

We thank the reviewer for this point. Related to this, we have strengthened the manuscript by including a figure which includes the data we have assembled on the number of courses of molnupiravir different countries have used, normalised by population. These limited data correlate well with the number of branches we see for a given region – though, as expected from factors such as differences in sampling, not wholly perfectly.

We agree that the reviewer’s calculation is interesting, however we think that the caveat of potential differences in sampling means that we would not be able to be confident in reporting such a figure. For example it is possible that people in care homes might be 10x more likely to be sequenced than those in the population as a whole, and also be much more likely to receive molnupiravir, which would affect these results. We believe that public health authorities with access to linkage data will be well-placed to lead future studies answering these questions. We believe some such projects are ongoing, and our work is likely to prompt further research in this area.

2. It may be useful to more explicitly contrast the hypermutation pattern (which is uniform across the genome and has $dN/dS = 1$) from that of hyperselected variants of concern, as in Omicron BA.1, which has a $dN/dS \gg 1$ with most coding changes in the S gene. That would strengthen the argument that the molnupiravir pattern is very different (and thus clearly discernible) from that of other means of natural selection.

We agree. Since initial submission we have performed additional work to look into selection effects and we now include this analysis. Briefly, we see a combination of two distinct effects. The first is the one that the reviewer notes. Long branches can arise in SARS-CoV-2 evolution without molnupiravir, but these branches, thought to result primarily in chronically-infected individuals, take a very particular form. They have, as the reviewer notes, a high dN/dS value, reflecting positive selection with a series of selective sweeps over time. When we perform the same analysis for our high G-to-A long branches, we find a substantially lower dN/dS value. As the reviewer suggests, this is likely to reflect the fact that the evolutionary forces are in this case dominated by the high rate of deleterious mutation, with selection acting primarily negatively to weed out such mutations.

But alongside this, we also see evidence of positive selection. We find that within high G-to-A branches non-synonymous mutations are more likely to occur in spike than in surrounding regions, and this was exacerbated when looking specifically at mutations which occurred repeatedly. These enrichments suggest positive selection for immune evasion, occurring alongside a dominant pattern of negative selection.

3. The exposition of the manuscript could be streamlined a little, but I commend the authors for conducting their analysis in a very short amount of time. The manuscript meanders somewhat between molnupiravir’s biochemical mechanism of actions, mutation patterns seen in data, epidemiology back to mutation patterns in the clinical trial.

We have now restructured the manuscript, as also suggested by Reviewer 2, to streamline the exposition and hope that this helps to make things clearer.

Referee #2

This is a really nice paper on an important topic. It has potentially substantial public health impact, and so I recommend publication as rapidly as possible (though I note and commend the authors for preprinting these findings alongside peer review).

Statistical tests are appropriate.

Detailed comments are below.

Major comments:

1. At present (presumably because this is the order of the work done by the authors) we get molnupiravir mutational theory, then the population data, then the direct examination of the drug's mutations from the AGILE data. I think the paper might flow better if the AGILE data is brought to the front, right after the intro on the theory, and then the population data are presented. Then we get: theoretical expectation of molnupiravir treatment, confirmation of it in trial data, and then search for that signature in the real world.

Thank you for these comments. We have reworked the manuscript in response to this point. We now introduce data from clinical trials first, using them to establish the particular significance of a high proportion of G-to-A mutations as a signal of molnupiravir usage. (We did experiment with introducing the concept of nucleotide contexts very early, but found that this interfered with flow.)

2. Following on from the point above, the authors used heuristics (20 subs to find long branches, then 10 subs with 25% G->A, 20% C->T, low transversions), again, I'm guessing because that's how they explored the GISAID data initially. I'm sure this gets at the key signal, but is there a more sophisticated way to find the molnupiravir-like branches, esp. knowing the trial data signature? e.g. there's a lot of 9 sub branches (Fig 4 suggests the length decays exponentially), and even a simple probabilistic model using the signatures might yield a lot more data to analyse.

We entirely understand the points that the reviewer is making, and that this heuristic may seem simplistic. We can assure reviewers that we've gone to considerable efforts to explore more complex models where we, for example, calculated the probability of observing a set of mutations under mutation-class rates from molnupiravir and those from typical SARS-CoV-2 evolution. We were surprised to find that these more complex models helped less than we had expected, for a few reasons.

Firstly, they did not create a substantial improvement in the sensitivity/specificity trade-off, although there were some changes. In part this may reflect the fact that a large number of mutations is itself a signature of molnupiravir usage and so considering lower mutation numbers adds less than one might expect.

Secondly, using a simple threshold-based approach allowed us to fairly perform analyses that would otherwise have been much more difficult. For example, as the reviewer identifies, we plotted the distributions of branch lengths of both "high G-to-A" and "normal" branches. If we had used a more sophisticated model which calculated a p-value based on each set of mutations, then this would have created a bias where we would be more likely to detect a high G-to-A branch when branch lengths were higher; this would make it more difficult to demonstrate the difference in branch length distribution due to the potential for confounding. We could have had a series of different definitions for different analyses, but this would make the paper harder to follow.

While the reviewer is right that there will be some molnupiravir events that lead to just 9 mutations, our data suggests this will not be a substantially greater number than those with 10 mutations (due to the flattish curve in the high G-to-A branch-length distribution).

The reviewer may also be suggesting the use of trinucleotide context data to detect molnupiravir based sequences. We definitely think this is a very valuable avenue, which we have pursued. Again there is a trade-off here. The most important outcome of our paper is to demonstrate conclusively that this effect actually exists in sequencing databases and can be reliably linked to molnupiravir. One way in which we achieve this is to first discover the sequences (without using trinucleotide context data) and then show that their trinucleotide contexts strongly match the contexts for known use of molnupiravir. If we used trinucleotide contexts to identify the sequences we would not be able to do this analysis without making a circular argument (or using a series of different definitions). We believe that incorporation of additional signals for detection of molnupiravir-associated branches will be valuable future work.

3. I think the authors need some precision about what group of "long branches" they are talking about at different points. For example on page 5 the authors examine spectra on "these branches". Is the antecedent the previous paragraph (v. long branches found specifically in Australia), or the aforementioned heuristic (10 subs with 25% G->A, 20% C->T, low transversions), which by definition will have G->A and C->T mutations, or something else? Similarly for Fig S7, which set of long branches is this?

We agree and in the revised manuscript have aimed to be clearer throughout.

4. Figure 3C needs some more explication for me. There does seem to be a relationship between number of sequences and number of long branches. Saying that the latter "could not be explained" by the former and then pointing to that graph needs more information. (Since it is log-log, it can be hard to realise how big an outlier, e.g. Australia and France, are). Actually without the logs, does it make the point more clearly?

We agree that this graph was difficult to interpret. We have: (1) added visual cues on the axes that this is a logarithmic graph (2) added a subsequent panel which normalises the axes against each other to provide a single figure (3) provided a supplementary figure (Fig. S1) which plots selected countries on linear axes. (If we use linear axes without faceting based on country, Australia dominates so much as to make other variation hard to interpret).

Minor comments:

- Abstract: "approach"

Thank you for pointing out this unfortunate typo.

- Abstract: "largely corresponds" is a bit weaselly. Be more precise?

We are now stronger in our claim ("matches"): this was made possible by the use of the Alteri et al. data which avoids an artifact that affected signatures in the AGILE trial.

- Figure S1 is nice, maybe pull it as a panel along Fig 1 2 into one place in main text?

We are grateful for this suggestion, and have implemented it.

- Figure 3A: "G-to-A ratio" Ratio of what? I think it is G-to-A to everything from the legend, but then I think "proportion" would be easier to understand than "ratio"? I think in other contexts "transition:transversion" ratio is common, but here again I think this is transitions:total (i.e. a proportion)

Agreed, and changed.

- Figure 3B: does this look any different if considered as a proportion, rather than absolute number? There are way more 2022 and 2021 sequences than 2020, as countries got their pipelines up and running.

We now provide this comparison in Fig S1. We agree this is an important comparison (the results have the same message).

- Figure S2: why does the blue line in the UK seem to have a different slope? (and/or why does the USA have no slope?)

In answer to the question: this may relate to differences in sequencing pipelines in different countries producing different technical effects. The expected appearance of such a graph in the absence of recombination and artifacts is a reasonably linear relationship, but of course recombination and sequencing artifacts both do occur. We decided that this figure caused more confusion than clarity and have removed it in the revised version in favour of simply reporting the result of a t-test.

- Figure 5A: The text says "closely related outgroups" of the long branch come from July 2022. Are there no dates for this branch itself?

We only have sequences and their metadata, and the branch is a computational feature that emerges from analysing them as a phylogenetic tree. We could attempt to date the tree using time-tree algorithms, but implicit to these tools is a model of evolutionary processes, and that model is likely to be different to the dynamics under molnupiravir treatment so we prefer to avoid this. What we aimed to note here is that the long branch, giving rise to many descendants sequenced in August, has sister short branches with descendants in July. This suggests a rapid accumulation of the mutations in the long branch.

- Figure 7A: Is it surprising (or not) that there are similar levels of C>T and G>A? Does the virus spend more or less time in the relevant phase of the replication cycle?

In our opinion this is not surprising. Molnupiravir is just as likely to be incorporated during negative-strand synthesis (creating a G-to-A mutation) as in positive-strand synthesis (creating a C-to-T mutation). This is in contrast to, say, APOBEC-based mutagenesis which involves access to genomic material at particular stages. In the revised manuscript we hope that we discuss these trends a bit more clearly, and colours in figures more clearly indicate the fact that contexts for C-to-T mutations match the reverse complement of contexts in G-to-A mutations, indicating this common origin.

- Figure 7B: It took me a couple of tries at the legend to understand this plot. At first I thought it was proportions of each mutation type among mutations, with each dot being I'm not sure what. Maybe be explicit in legend: "each dot is a trinucleotide context". Could even call out one dot in the plot "TGT or ACA", or similar?

We are grateful for this suggestion and now do so throughout the manuscript.

- Figure 8C: there are more of nearly all kinds of mutations in the treatment arm. Any idea why? Direct molnupiravir effect not in the canonical places from Fig 2? Indirect effect of initial mutations on weaker error control? Something else?

In the previous version of the paper we focused on the AGILE dataset as the source for this "ground truth" data for molnupiravir. However there appears to be a G-to-T sequencing artifact in the AGILE dataset, which we now discuss in more detail in the methods section. This technical artifact affects both the placebo and the molnupiravir arms, but especially the molnupiravir arm (likely to be due to the reduced viral loads in the molnupiravir arm).

We now avoid this effect entirely by focusing the primary analysis on a newly released alternative dataset from Alteri et al., and use the AGILE dataset only for further confir-

mation. In the revised dataset, the elevated signature is much more specific to transition mutations.

- Figure S4 is not referenced until the methods, and Figure S5 is not referenced anywhere.

We thank the reviewer for noting this and now reference all figures.

- Methods: the metadata adding heuristics seem like they have some problematic edge cases (e.g. at calendar breaks). I doubt it makes any difference, but the authors might want to comment.

This is a perceptive point. As the reviewer predicted this does make little difference, and it has certain advantages over a time-tree based approach (because it is not biased by the number of mutations observed), and also helps to exclude branches that represent shared sequencing artifacts. We have added a sentence commenting that this does not exclude many nodes.

- I found the layout of the PDF absolutely delightful to review!

Thank you!

Referee #3

Sanderson et al present a retrospective computational phylogenetic analysis of over 13 million SARS-CoV-2 genomes from GISAID. They identify a set of unusually mutated sequences, bearing a particular signature enriched for G>A and C>T mutations, and propose that these sets may have been the result of intra-host evolution subject to (ineffective) applications of a mutagenising agent molnupiravir. There are three lines of evidence offered to justify this connection

- 1). That these mutational signatures are consistent with what would be expected from the molecular basis of action of molnupiravir.
- 2). That these mutations were contemporaneous with the approval and large-scale administration of molnupiravir in some (but not all countries), and that the set of sequences bearing these mutations was enriched for older patients (where such data are available).
- 3). That these mutations are largely (but not 100%) consistent with a study that performed direct sequencing of subjects enrolled in an RCT examining the efficacy of molnupiravir.

This is an intriguing study which is innovative and insightful in the following ways

- 1). The use of a large scale genomic database and "big-data" phylogenetic tools to uncover previously under-appreciated but biologically important patterns.
- 2). Indirect evidence of unexpected and potentially highly undesirable cases of drug-escape and accelerated evolution of RNA viruses, including evidence of onward transmission.

The second point above is of broad interest because it showcases how unanticipated effects of treatment with "unique" therapeutic agents may lead to the unexpected diversity of the targeted pathogen.

There are, however, several issues with the study.

- 1). The conclusions of the study rest on two big assumptions.
 - (a) Since we have limited knowledge of what "in the field" molnupiravir-induced evolution would look like, we must accept that G>A and C>T "hypermutations" are in fact attributable to molnupiravir-driven mutagenesis. I think the authors could have done considerably more here to further support this claim
 - (b) An even more severe limitations is that there are no meta-data on treatment histories in GISAID, hence we don't know which of the individuals with hyper-mutated sequences have been exposed to molnupiravir
 - (c) Because GISAID does not readily make underlying NGS data available, quality control and validation of such hyper-mutated genomes (which would be requisite to rule out

batch artifacts, and other technical issues) is impractical. Thus one is left to wonder (especially with cases like shown in Fig 6C with > 100 mutations) if these sequences are trustworthy.

We thank the reviewer for pointing out that our manuscript did not include an analysis of treatment records. In the revised manuscript we have partnered with UKHSA to perform an analysis of treatment records, which find a >500 -fold enrichment for molnupiravir-prescription in the branches that we identify here. We hope that this provides considerably more confidence to the conclusions drawn.

We understand the reviewer's initial concerns about mutations with >100 mutations. We agree that upon seeing an arbitrary sequence with 100 mutations, our own instinct would be to expect a technical artifact. However, a number of factors lead us to conclude the sequence is reliable in the cases we identify. High G-to-A sequences with hundreds of mutations are all from Australia, which has performed high quality sequencing during the pandemic with careful quality-control. For instance, in one case of one noted molnupiravir-induced Australian lineage, the researchers were able to go back and re-sequence all samples, and confirm in the pango-designations repo that sequencing was consistent, which they reported in <https://github.com/cov-lineages/pango-designation/issues/1286#issuecomment-1303013648>. The reviewer might be concerned about a technical effect in a laboratory in Australia driving this effect but: sequences come from multiple Australian laboratories, and Australia has also used more molnupiravir per capita than any country for which we could find data, providing an explanation for this enrichment. These errors are not random, they fall into molnupiravir's distribution of mutation classes, and also on a per-sequence level have trinucleotide contexts which match those of known molnupiravir use (data available on request). Thus we are confident that these sequences are genuine.

(d) I am quite concerned that the authors did not really address the G>T mutation which IS enriched in the trial data, but NOT enriched in their GISAID clusters.

The reviewer is right to make this point about G-to-T mutations. We had not delved deeply into this G-to-T signature in the AGILE data because we were relatively confident that it represented a technical artifact, on the basis of several *in vitro* studies, and the fact that even the placebo group had highly elevated G-to-T substitution rates. We agree that providing further evidence that patients treated with molnupiravir do not develop this mutation type is an important step. In the revised manuscript we have analysed an alternative recently published clinical study that provided longitudinal data for patients treated with molnupiravir (Alteri et al, 2022). In this study we again see increases in G-to-A and C-to-T mutations, as well as A-G and T-to-C, but we do not see the G-to-T signature. This supports results seen previously in other studies, that molnupiravir specifically induces transitions. This change has also resulted in our correlations between trial data and long G-to-A branches within G-to-A mutation and C-to-T mutation contexts being much stronger ($R=0.97$, $R=0.9$).

These issues makes causality impossible to establish, and muddies even associations, because we don't know the true "labels" for the sequences. While some of the indirect evidence (timing, association of molnupiravir availability with hypermutations) is suggestive, it is not sufficient to draw any strong conclusions.

As discussed above, the revised manuscript contains extensive further evidence for a link to molnupiravir:

- Analysis of treatment records showing a >500 -fold enrichment in molnupiravir use in high G-to-A branches in England
- A new source of ground-truth data for the molnupiravir spectrum, which avoids a technical artifact and results in a near-perfect match with the high G-to-A spectrum from the mutation annotated tree.

In addition to the correlations, both in time and place, with the use of molnupiravir, evidence for the action of a mutagen in these branches also comes from the fact that branches with this signature very often involve the emergence of 20 or more mutations over a very short period of time (1-2 months), which is extremely atypical of SARS-CoV-2 evolution. Our revised work contains an appendix of trees to make this effect clearer.

2). Even if the discovered hyper-mutated clusters are attributable to molnupiravir, the manuscript does nothing at all to attempt to establish the significance of this. Are these mutations in any way likely to create altered viral phenotypes with a potential selective advantage (including resistance to molnupiravir)? Seeing how effectively all of the detected events are dead-ends, this is not very likely.

Throughout this work we have been intentionally careful about the conclusions that we draw. Our work establishes conclusively that viruses heavily mutated by the action of molnupiravir can be fit enough both to survive in the patient, and (sometimes) to be transmitted. When molnupiravir was being considered for authorisation, data was presented that showed that no infectious virus could be detected after treatment, and so our data provide a new perspective.

The reviewer is correct that our first manuscript did not include analysis of the action of selection on these genomes. In the revised manuscript we provide an analysis of the mutations that recur most often in these molnupiravir-associated branches. We observe that the most recurrent mutations in these branches include spike mutations that occur commonly in chronically infected patients and have been implicated in immune evasion. These include a mutation that gives rise to resistance to a monoclonal antibody therapy.

3). The manuscript needs to improve its statistical rigour. This takes many forms (a). Lack of any formal statistical testing for many visual associations (e.g. Figure 3 C and D) We thank the reviewer for raising this point and have added p-values for these associations in the figure captions (now Fig 2F, Fig2G). (b). Seemingly arbitrary definitions of mutational signatures in terms of G/A C/T ratios etc

We discuss this point a little in response to reviewer 2 above. We believe that clearly communicable thresholds are a valid and effective way of defining this criterion. We have experimented with probabilistic definitions but found that these did not provide a net improvement because they limited the downstream analyses that could be performed because the likelihood of detection varied with branch length.

(c). Lack of any kind of explicit null. It would be highly informative to estimate, using for example some available models of mutational preferences/biases for SARS-CoV-2, how likely are "natural" processes to generate particular features mutational signatures. As it is, there is no quantitative basis to evaluate "deviations from expected values"

We hope that the added linkage analysis to patient treatment records addresses the link to molnupiravir more explicitly.

One issue we observed with, for example, chi-squared approaches to looking at deviation of mutation classes from those under the typical Omicron spectrum is that there may not be a single well-defined "null" value. With some similar approaches we found that we included clusters from SARS-CoV-2 in deer, which also appear to have a (subtly different) change in their mutation-class distribution, likely driven by APOBEC mutagenesis. We would argue that in some sense we use the best "null" here, the natural null of sequences from 2021, before molnupiravir, and sequences from countries where molnupiravir was not used. An advantage of this approach is that it accounts for the fact that there may not be a single distribution of mutation classes from which mutations are drawn, even in the absence of molnupiravir. However in light of the suggestion below for sensitivity analyses we have added a sentence describing the expected sensitivity and specificity if

mutations were indeed simulated as drawn from the naive or molnupiravir mutation class distributions. This yields a specificity of 99.7% and a sensitivity of 51

(d). Missing data is a major issue here that is not addressed; for example Figure 3D comparison implies that age data are missing entirely at random.

We did not intend to suggest that data are missing entirely at random. We entirely agree that data will be missing: firstly only a small number of SARS-CoV-2 infections are sequenced, so some data will be missing entirely (the dataset likely has a general bias towards molnupiravir-treated individuals). Secondly, there may be age-specific differences in metadata availability. We don't believe this causes a significant issue to our analysis. The claim we are making is that high G-to-A branches on average occur in older individuals than other branches. For this effect to be confounded, missingness of metadata would have to be connected to the presence or absence of the high G-to-A criterion and this seems unlikely (unless the effect is mediated by molnupiravir or another age-specific treatment). However we have now made a change to specifically flag potential missingness in the figure caption.

Specific suggestions on further analyses.

1. Mutational signatures (i.e. counts of nucleotide mutations) form a very crude measure. Evolution due to molnupiravir might leave other, detectable signatures (a). What is the distribution of synonymous and non-synonymous mutations? Molnupiravir might induce a "neutral-like" evolutionary pattern. Alternatively, in order to survive, the putative molnupiravir sequences you see could be enriched for synonymous mutations. If you look at the mutations, what do they look like? Do you see any in RBD that have any significance (e.g. nAb or immune escape) and that have not been seen in circulating isolates? Do you see unusual indels in these sequences (not expected to be caused by molnupiravir)

This is an important question which we are pleased to address. We have added data on dN/dS values. We find that high G-to-A branches have a lower dN/dS value than other long branches (which have high dN/dS, likely driven by chronic mutations). This is likely in part to reflect purifying selection against molnupiravir-induced mutations.

We also analyse the locations at which mutations occur. Many of the most recurrent mutations are indeed regularly occurring mutations in chronic individuals, associated with immune escape, and the high enrichment for recurrent mutations occurs in spike.

USHER trees do not model indels and so these are excluded from the analysis.

(b). What do the spatial distribution of mutations look like? If randomly generated, they should NOT look like the mutations driving variant evolution, i.e. be different from others

We now provide the spatial distribution of mutations in a figure. Synonymous mutations show a mostly even distribution, while non-synonymous mutations show a small local maximum in spike, and the most recurrent mutations a larger peak in *spike*. As we discuss above and in the manuscript we interpret the dN/dS values as suggesting a general dominance of purifying selection, with the distribution of mutations indicating some positive selection on top of this.

(c). What about the other long branches (Figure 4)? They are not discussed at all.

We are not trying to claim that all long branches in SARS-CoV-2 evolution are driven by molnupiravir: we know that long branches can arise for example in chronic infections. We now look at this a bit more explicitly by showing what proportion of long branches satisfy our high G-to-A criterion, faceted by country, and by discussing chronic infections in the context of the selection analysis.

2. Sensitivity analysis with respect to how these mutational signatures are defined.

We are grateful for the suggestion. As discussed above we have now added an analysis of simulated sensitivity and specificity under the assumption that mutations are simply drawn from a mutation class distribution that is either naive or molnupiravir.

3. More careful attention to interpretation and discussion. For example, assuming that a molnupiravir lineage does survive and even spread, what do you expect to happen?

We do not know what will happen if a molnupiravir-derived lineage survives and spreads. Each and every molnupiravir-derived lineage will be different and their effects will differ depending on the mutations they carry. We aimed to be careful in our Discussion to clearly communicate that it is difficult to make predictions about the impact of molnupiravir on the probability of evolution of new variants of concern. We offer a range of points in this discussion, from both sides of the debate. Our revised paper does provide more evidence on these points, both from our new selection analyses and from including further information from clinical trials.

Figure 5A for example shows some odd patterns which may look like reversions (LATER sequences are closer to the MRCA than EARLIER sequences). What other processes could have given rise to these patterns?

We agree with the reviewer that this phylogenetic analysis is not able to definitively resolve the dynamics downstream of this molnupiravir-associated event (its key role is to resolve the event itself). The reason that three sequences with dates of 22 Aug and 30 Aug are placed close to the MRCA is that they have missing coverage in some regions, and so the algorithm cannot exclude that they have the same sequence as the MRCA at these sites. We have added a sentence to this effect in the caption.

Can you better correlate the intensity of treatment with the number of clusters?

We now present the data dichotomised by approval much more clearly in Fig 2F, normalising for total genomes.

We have also attempted to assemble as much data as possible on how the intensity of treatment differed by country which yielded 4 data points (shown in brackets in Fig 2F) – and also partial data from the USA. These correlations point in the right direction, with Australia having the most intense treatment and the most clusters, and the UK little treatment among approving countries, and few clusters. But the data are scarce and we don't overstress this point.

Do different SC2 lineages experience similar/different levels of molnupiravir-induced mutagenesis?

Answering this question convincingly from our data would be very difficult due to confounding from the fact that lineages change over time and molnupiravir usage also changes over time. In vitro studies would be well-placed to help with this. To our knowledge, differences have not been reported.

Reviewer Reports on the First Revision:

Referees' comments:

Referee #1 (Remarks to the Author):

The authors have fully addressed all my concerns. The reasoning and evidence is convincing and the presentation is clear. I have no further comments.

Referee #2 (Remarks to the Author):

The authors have done excellent revisions that address comments from me and the other reviewers. I think the story flows nicely now, and builds to quite a climax. I was inclined to agree with reviewer 1 that the authors already had a smoking gun, but wow, here's the gun, the shell casings and a ballistics match:

"This analysis found that 31% of clades descending from a high G-to-A branch involved at least one person prescribed with molnupiravir, representing a more than 500-fold enrichment over the proportion expected for randomly selected sequences (the overall rate of molnupiravir prescription in sequenced individuals in England from 2022 is 0.043%)."

My one remaining request is that maybe they find a way to describe, either in methods or main, that they tried more elaborate approaches than the heuristic for branch finding, but they weren't any better (and would've made other analyses impossible/harder, as they pointed out in the response letter).

A few minor comments:

- worth pointing out that the PANORAMIC trial was in (almost entirely) vaccinated individuals in the intro?

- Fig 2F could we have 10^{-3} instead of $1e-03$ etc

- Methods p9 "which contained longitudinal both individuals treated" missing a "data", or something like that?

Referee #3 (Remarks to the Author):

The revised study by Sanderson et al is significantly improved compared to the previous submission.

The authors conducted several new analyses, and made use of the unique UK-based database (Blueteq) to enhance the plausible causal link between long G->A (and C->T) branches that are consistent with the molecular action of molnupiravir. A better focus on transmission clusters of molnupiravir-associated variants is also very compelling.

The authors have addressed most of my concerns, and those of the other reviewers. Text and figure clarity have also been greatly improved; this paper will now be readily followed by the general scientific readership of Nature.

Overall, I am supportive of seeing this published in Nature, because the study is innovative in its use of "pandemic-scale" genomic data, and impactful, in that it clearly demonstrates that among many failures of policy and implementation of the COVID-19 response, the rapid approval and wide-spread use of molnupiravir was not beneficial to disease control and treatment, yet it left an unexpected (or perhaps, entirely expected) and potentially dangerous "anthropogenic" mutational signature in the genomes. I would strenuously encourage the authors to strengthen their conclusion: MPV clearly mutagenizes the virus in UNEXPECTED ways (that the original safety trials missed). This virus is transmissible, and strongly departs from normal evolutionary trajectories of the SC-2. Combined with reported (but ignored) long term dangers of DNA-analogs to the integrity of the host DNA (<https://pubmed.ncbi.nlm.nih.gov/33961695/>), the wide-spread use of MPV, given its very limited (if any) clinical efficacy, represents a failure of regulatory bodies, and an major lesson for future drug approval and safety evaluation.

My remaining comments can be considered minor; but I suggest they be addressed prior to publication to ensure maximal technical rigor and reproducibility. Some of my comments are more discursive and simply bring up questions that the manuscript did not fully address.

-- The G/A mutational signature definition remains sloppy. For example, 95% of 10 mutations is 9.5 mutations. Does it mean that all 10 mutations have to be transitions? I have no earthly idea, based on the text provided, how the sensitivity/specificity simulation was done. Presumably one just simulates X mutations using the conditional (triplet-based) distribution sited, and sees how often you meet the criteria? Why does the sensitivity remain low (~50%)? Which patterns are you missing?

-- For patient age estimation, assuming that the parent age is simply the mean of its descendants is a rather strong assumption which could bias the results. The authors might want to acknowledge that.

-- Figure 2B shows a curious artefact of data compositionally (sums to 1): while the expected G:A mutation surfeit is patently obvious, one also observes a relative paucity of A:T, C:A, etc. The molecular mechanism of MPV does not SUPPRESS those mutations, but whatever accrues to G:A must be taken from other mutation types.

-- Figure 3D; I would suggest adding some sort of uncertainty quantification on proportions (e.g. using multinomial CI). Would help interpret deviations from the diagonal and see if any proportions are significantly different between the two estimates.

-- Also figure 3D; when comparing proportions, it is not appropriate to use regression, as it does account for compositionally of data. You can use `prop.test` in R, or goodness-of-fit tests, or any test that is designed to compare multinomial proportions.

-- Figure 4C; without additional verification of raw data, how can we be

sure that this is not a technical artifact? A 4+ month gap to closest relatives seems odd as well.

-- Figure 4; since there is a lot of white space on the long branches, would be informative to add more annotations on the bubbles; not just break down mutations by nucleotides, but perhaps indicate individual Spike mutations, or dN/dS?

-- Figure 4A; when multiple waves of Tx follow an MPV branch, what mutations accrue on other internal branches (e.g. the parent of EPI_ISL_15081441 and EPI_ISL_15081620)? Would you expect these to be "nominal", i.e. like in other Tx (non MPV induced)?

-- I recognize that the authors have added a dN/dS (selection) analysis because of an explicit request. However, the way it was done creates more confusion than clarity. Firstly, the way dN/dS is computed is rather strange. The authors re-implement a simplified version of the 1986 Nei-Gojobori dN/dS estimator. They calculate dN/dS normalization based on the reference genome, and not the ancestral/source genome (which is available from BTE), do not account for any mutational biases (which are very strong for SC-2 and are very likely to bias the results), and obtain a genome-wide (instead of, e.g. gene-wide) estimates. There is no statistical testing performed on point estimates (NG86 has asymptotics, bootstrap can be used as well). A point estimate of genome-wide dN/dS=0.73 does not indicate positive selection. My suggestion would be to completely remove this section, but keep Figure 5C and 5D.

-- Figure 5C/5D brings up an very interesting pattern in Spike and should be discussed in greater detail. There clearly is more non-synonymous and especially recurrent non-synonymous, substitutions in Spike for the MPV-associate branches. Why? Are these mutations occurring in preferred MPV contexts (TGT, TGC)? Is there enough time during a typical MPV-treatment to "fix" additional beneficial mutations following the introduction of MPV-induced errors? Is there any difference in Spike mutations between those sequences which transmit onward, and those which are dead ends? I suppose the "null" explanation is that sometimes (given the large number of replicates), MPV mutations just happen to hit the right targets in Spike. Yet, this could be tested by simulating mutations according to the predicted/observed distribution (conditioned on the number), and seeing how often you get recurrent or constellations of recurrent mutations.

-- Along those lines, one might expect spatial uniformity of MPV action, i.e. the fractions of mutations falling into a gene should be roughly proportional to its length corrected for mutation preference biases. Do we see this? If not, would indicate some sort of selection or deviation from assumptions.

-- The R notebook in the attendant repository is, strangely, a PDF file. Makes reuse unnecessarily painful.

Author Rebuttals to First Revision:

Referee #2 (Remarks to the Author):

The authors have done excellent revisions that address comments from me and the other reviewers. I think the story flows nicely now, and builds to quite a climax. I was inclined to agree with reviewer 1 that the authors already had a smoking gun, but wow, here's the gun, the shell casings and a ballistics match:

"This analysis found that 31% of clades descending from a high G-to-A branch involved at least one person prescribed with molnupiravir, representing a more than 500-fold enrichment over the proportion expected for randomly selected sequences (the overall rate of molnupiravir prescription in sequenced individuals in England from 2022 is 0.043%)."

We thank the reviewers for their kind and poetic appraisal.

My one remaining request is that maybe they find a way to describe, either in methods or main, that they tried more elaborate approaches than the heuristic for branch finding, but they weren't any better (and would've made other analyses impossible/harder, as they pointed out in the response letter).

We agree that this is useful, and have added a section in the limitations section including these points. (The limitations section has had to move to Methods due to the instructions to authors requesting a very brief discussion section in the main text.)

A few minor comments:

- worth pointing out that the PANORAMIC trial was in (almost entirely) vaccinated individuals in the intro?

Thank you for this suggestion, which we have added.

- Fig 2F could we have 10^{-3} instead of $1e-03$ etc

We have implemented this.

- Methods p9 "which contained longitudinal both individuals treated" missing a "data", or something like that?

Thank you! We have fixed this.

Referee #3 (Remarks to the Author):

The revised study by Sanderson et al is significantly improved compared to the previous submission.

The authors conducted several new analyses, and made use of the unique UK-based database (Blueteq) to enhance the plausible causal link between long G->A (and C->T) branches that are consistent with the molecular action of molnupiravir. A better focus on transmission clusters of molnupiravir-associated variants is also very compelling.

The authors have addressed most of my concerns, and those of the other reviewers. Text and figure clarity have also been greatly improved; this paper will now be readily followed by the general scientific readership of Nature.

Overall, I am supportive of seeing this published in Nature, because the study is innovative in its use of "pandemic-scale" genomic data, and impactful, in that it clearly demonstrates that among many failures of policy and implementation of the COVID-19 response, the rapid approval and wide-spread use of molnupiravir was not beneficial to disease control and treatment, yet it left an unexpected (or perhaps, entirely

expected) and potentially dangerous "anthropogenic" mutational signature in the genomes. I would strenuously encourage the authors to strengthen their conclusion: MPV clearly mutagenizes the virus in UNEXPECTED ways (that the original safety trials missed). This virus is transmissible, and strongly departs from normal evolutionary trajectories of the SC-2. Combined with reported (but ignored) long term dangers of DNA-analogs to the integrity of the host DNA (<https://pubmed.ncbi.nlm.nih.gov/33961695/>), the wide-spread use of MPV, given its very limited (if any) clinical efficacy, represents a failure of regulatory bodies, and an major lesson for future drug approval and safety evaluation.

We thank the reviewer for assessing our work again in light of the revisions, and for their supportive words.

We also thank them for the suggestion that our conclusions could be strengthened. We have a careful balance to strike in this paper. We have identified an important pattern in global sequences, and we aim here to provide conclusive evidence that this is due to the action of molnupiravir, which is causing mutations that persist long enough to be captured in sequencing databases, sometimes with very large numbers of fixed mutations, and on occasion with strong evidence that transmission of molnupiravir-derived clades has occurred. We understand the point the reviewer is making that our work adds to a broader potential negative narrative around molnupiravir. We already reference the negative results from the PANORAMIC trial, and in the last revision we added a pointer in the discussion to that trial's perhaps under-appreciated observation that viral load was somewhat increased in day 14 in molnupiravir-treated individuals as compared to controls. In general, we want to stick in our paper largely to what we can directly show with our analyses, leaving any broader synthesis to others.

All that said, we have made changes to the introduction and discussion which may be of some comfort to the reviewer.

Most importantly, we agree with the reviewer that it is important to highlight that our results suggest that use of molnupiravir causes SARS-CoV-2 to differ from its normal evolutionary trajectory. In the discussion we now point to the fact that, as the reviewer suggests, molnupiravir's divergent mutation spectrum means that the virus may be exploring genotypes it would not normally encounter under typical evolutionary conditions. In response to *Nature* formatting requirements we have generally pared-down the discussion, and believe this may as a side-effect have made it slightly harder-hitting.

We have added a mention of the potential for mutagenic activity in the host to the introduction, referencing a review on the topic, which we agree is important context.

We have also highlighted in the discussion that molnupiravir is rare even among nucleoside analogs in acting purely through mutagenesis, as opposed to some other analogs that function entirely or primarily through chain-termination.

My remaining comments can be considered minor; but I suggest they be addressed prior to publication to ensure maximal technical rigor and reproducibility. Some of my comments are more discursive and simply bring up questions that the manuscript did not fully address.

-- The G/A mutational signature definition remains sloppy. For example, 95% of 10 mutations is 9.5 mutations. Does it mean that all 10 mutations have to be transitions? I have no earthly idea, based on the text provided, how the sensitivity/specificity simulation was done. Presumably one just simulates X mutations using the conditional (triplet-based) distribution sited, and sees how often you meet the criteria? Why does the sensitivity remain low (~50%)? Which patterns are you missing?

We thank the reviewer for these points. We have amended the paper to clarify our definition of high G-to-A sites, and in doing so we have uncovered and fixed a small but important inconsistency. In the revised version we had dropped this threshold to a more permissive 90% (which was noted in the caption of Fig. 2), but had failed to update this both in the main text, and in the calculations to simulate sensitivities. We regret these inconsistencies, and have now made sure they cannot recur by hard-coding these shared variables. These updates increase our simulated sensitivities to 47% (branch length 10), 71% (branch length 15) and 63% (branch length 20). We apologise for this error, and thank the reviewer for the chance to check our work here. Our results are qualitatively unchanged, but this is nonetheless an important correction.

The reviewer's interpretation of the mathematics is correct. In the new case 90% of 10 mutations is 9 mutations, and we use a "strictly greater than" for this criterion, so all mutations must be transitions at 10 mutations (for 11 mutations, 1 transversion is accepted). Transitions are more common than transversions in typical SARS-CoV-2 evolution even in the absence of molnupiravir, and so the penalising of transversions does not end up being overly onerous in our opinion. Ultimately this is all a matter of trade-offs though. The key aim of this paper is to robustly demonstrate that these molnupiravir-derived mutational events truly exist, and we achieve this in part through correlational analyses of timing and geographic location. Because molnupiravir treatment is a rare event, we need very high specificities to show the increase due to molnupiravir clearly, and this limits our ability to achieve maximal sensitivity.

We have expanded the description of our simulated sensitivity and specificity, which we reproduce below. (This supplements the availability of this simulation in the code).

We defined "high G-to-A branches" as those with at least 10 mutations, of which >90% were transitions and >25% were specifically G-to-A mutations, with >20% C-to-T. Such a threshold appeared to yield very high specificity, as judged by the ability to detect marked changes in the rate of a rare event (molnupiravir treatment) over time. We also created simulated measures of sensitivity and specificity using the distribution of mutation types from

Ruis et al. and Alteri et al. We performed these calculations for different branch lengths (n) from 10 to 20. In each case we performed 10,000 draws of n mutations from each of the naive and molnupiravir-associated mutational class distributions. We then assessed what proportion of these draws satisfied our criteria defined above. In the case of the molnupiravir-associated class distribution, this proportion represented the sensitivity. In the case of the typical-BA.1 distribution, this proportion represented 1 - specificity.

-- For patient age estimation, assuming that the parent age is simply the mean of its descendants is a rather strong assumption which could bias the results. The authors might want to acknowledge that.

We definitely want it to be clear that ages are assigned using a simple heuristic in cases where there are multiple descendants of a node. In many cases nodes do have single descendants, and those cases will be unaffected by this effect, but where nodes have multiple descendants we need some way to propagate age data to parents (failing to do so could itself introduce biases), and so we adopt this simple procedure to do so. To be sure that the link between the high G-to-A signature and age that we observe is not due to a bias, we have now repeated the age analysis but filtering to nodes with only a single descendant which are unaffected by this heuristic. The mean for the *Other* group is unchanged at 43, and the mean for the high G-to-A group increases from 60 to 62, slightly increasing the strength of the effect from that reported in our manuscript, and confirming that the effect we observe is not due to a bias. (These results are provided simply for the purposes of review). We have also now added a mention of the heuristic to the figure caption for clarity.

-- Figure 2B shows a curious artefact of data compositionally (sums to 1): while the expected G:A mutation surfeit is patently obvious, one also observes a relative paucity of A:T, C:A, etc. The molecular mechanism of MPV does not SUPPRESS those mutations, but whatever accrues to G:A must be taken from other mutation types.

The reviewer's interpretation is absolutely correct. We have added a sentence to the figure caption to clarify this.

"These are ratios of proportions and so the apparent reduction in transversions does not require an absolute decrease in the number of transversions, but can instead be caused by the increased number of transitions."

-- Figure 3D; I would suggest adding some sort of uncertainty quantification on proportions (e.g. using multinomial CI). Would help interpret deviations from the diagonal and see if any proportions are significantly different between the two estimates.

We are grateful for this advice, but believe that overall this plot might become overloaded and difficult to read with lines used to indicate uncertainty given that lines are already used to link points to their context label.

-- Also figure 3D; when comparing proportions, it is not appropriate to

use regression, as it does account for compositionally of data. You can use `prop.test` in R, or goodness-of-fit tests, or any test that is designed to compare multinomial proportions.

We thank the reviewer for highlighting this and have shifted this approach to report cosine similarity, which is a measure widely used in mutation spectrum analysis.

As best we could tell, `prop.test` appears to expect binomial situations, and more generally, we are not aiming here to perform null-hypothesis significance testing against a null that these distributions are exactly the same: we fear that such an approach could either fail to reject this null hypothesis on the basis of having too little data, though the distributions might not be the same, or on the other hand reject it on the basis of the distributions being very subtly different, and having a very large n to detect that subtle difference, although that subtle difference could be due to, say, one setting being a clinical trial and the other being "in the wild" (the latter is more likely to involve also background mutations). Instead we aim to assess the overall similarity of the two spectra, which cosine similarity measures.

-- Figure 4C; without additional verification of raw data, how can we be sure that this is not a technical artifact? A 4+ month gap to closest relatives seems odd as well.

We are grateful for the opportunity to address this further, and have added two additional aspects to the manuscript to address this. We will firstly quote the points made on this point in our previous rebuttal, then move on to the novel aspects in the revised paper.

"We understand the reviewer's initial concerns about mutations with >100 mutations. We agree that upon seeing an arbitrary sequence with 100 mutations, our own instinct would be to expect a technical artifact. However, a number of factors lead us to conclude the sequence is reliable in the cases we identify. High G-to-A sequences with hundreds of mutations are all from Australia, which has performed high quality sequencing during the pandemic with careful quality-control. For instance, in one case of one noted molnupiravir-induced Australian lineage, the researchers were able to go back and resequence all samples, and confirm in the pango-designations repo that sequencing was consistent, which they reported in <https://github.com/cov-lineages/pango-designation/issues/1286#issuecomment-1303013648>. The reviewer might be concerned about a technical effect in a laboratory in Australia driving this effect but: sequences come from multiple Australian laboratories, and Australia has also used more molnupiravir per capita than any country for which we could find data, providing an explanation for this enrichment. These errors are not random, they fall into molnupiravir's distribution of mutation classes, and also on a per-sequence level have trinucleotide contexts which match those of known molnupiravir use (data available on request). Thus we are confident that these sequences are genuine."

In the newly revised manuscript we have expanded on this to further demonstrate that this sequence is truly due to molnupiravir.

We believe the reviewer is now convinced that the *overall* mutational pattern we describe in this paper is in general due to molnupiravir (based on the correlational data, linkage analysis,

and mutation spectrum analysis). However they are concerned that there is not sufficient evidence to claim that this particular sequence, with >100 mutations is due to molnupiravir rather than to a technical artifact. We now supply this evidence. This sequence was identified as a candidate for being molnupiravir-related purely on the basis of the *types* of mutations observed, which are enriched for G-to-A and C-to-T mutations. We have observed in the paper that these types of mutations are related to molnupiravir, but we have also observed an independent effect: that molnupiravir especially drives mutations in particular contexts. For example, G-to-A mutations are especially common in TGT contexts. We can examine the contexts of the mutations in this branch to see if they share the contextual preferences observed for molnupiravir.

For each transition mutation class we evaluate the likelihood of observing the pattern of contextual mutations seen here under either the molnupiravir mutation spectrum, or the BA.1 spectrum. Each transition mutation type (G-to-A, C-to-T, A-to-G, T-to-C) independently shows a greater likelihood of the results coming from the molnupiravir spectrum, and the combined Bayes factor is >1e10. There is no reason that a technical artifact would cause an effect that matches molnupiravir in this way.

Secondly, in the weeks after this sequence was first included in our manuscript, further closely related sequences (EPI_ISL_16315710, EPI_ISL_16639468) from the same hospital were uploaded. These sequences share a large number of the private mutations of EPI_ISL_16191277, but not all of them, and have their own sets of unique mutations. This is a pattern commonly observed when a single patient with a chronic infection is sampled multiple times. We believe that this both provides evidence of the reality of these mutations, and also that it is important to mention this given that our previous description of a "singleton sequence" has become technically inaccurate. We therefore now mention these effects in the figure caption.

-- Figure 4; since there is a lot of white space on the long branches, would be informative to add more annotations on the bubbles; not just break down mutations by nucleotides, but perhaps indicate individual Spike mutations, or dN/dS?

We thank the reviewer for this point and have added an annotation which lists coding changes in Spike in each case.

-- Figure 4A; when multiple waves of Tx follow an MPV branch, what mutations accrue on other internal branches (e.g. the parent of EPI_ISL_15081441 and EPI_ISL_15081620)? Would you expect these to be "nominal", i.e. like in other Tx (non MPV induced)?

This is a great question. Ultimately we would certainly expect it to be entirely *possible* to see non-molnupiravir-induced mutations following a molnupiravir-associated event as in any infection and transmission mutations of any type could occur. In the full set of trees we provide in the supplement, when individual cases are inspected (on usher.bio, enter an EPI_ISL id to query and examining the local tree for mutations), there are a number of cases

where transversion mutations follow the main molnupiravir-associated event, e.g. A-to-C and G-to-T mutations in the Austrian cluster containing EPI_ISL_14355571.

That said, in the specific case that the reviewer highlights, the parent of these two sequences contains a G-to-A mutation which seems likely to be molnupiravir-derived, and there are several other transitional mutations in this cluster subsequent to the main branch, which could be due to the action of molnupiravir. This is not an uncommon pattern in these molnupiravir-associated clusters. As the reviewer may be alluding to, this could represent further transmissions of subtly different genotypes from a mixed population in the molnupiravir-treated index case.

-- I recognize that the authors have added a dN/dS (selection) analysis because of an explicit request. However, the way it was done creates more confusion than clarity. Firstly, the way dN/dS is computed is rather strange. The authors re-implement a simplified version of the 1986 Nei-Gojobori dN/dS estimator. They calculate dN/dS normalization based on the reference genome, and not the ancestral/source genome (which is available from BTE), do not account for any mutational biases (which are very strong for SC-2 and are very likely to bias the results), and obtain a genome-wide (instead of, e.g. gene-wide) estimates. There is no statistical testing performed on point estimates (NG86 has asymptotics, bootstrap can be used as well). A point estimate of genome-wide dN/dS=0.73 does not indicate positive selection. My suggestion would be to completely remove this section, but keep Figure 5C and 5D.

We thank the reviewer for these points. We did not aim to suggest that a value of 0.73 indicated overall positive selection. All dN/dS values capture a combination of various selective forces, and we were aiming to suggest that the positive delta from 0.45 to 0.73 could be explained by an increase in positive selection in chronic infections relative to typical evolution. As the reviewer points out, our treatment here was simple: we used BTE to assess what proportion of mutations are synonymous/non-synonymous under different conditions and then normalised to the number of sites available for single-nucleotide-based synonymous or non-synonymous mutations to estimate dN/dS.

We think that there is value in examining the effects of mutations created by molnupiravir, but we acknowledge that the reviewer feels that our treatment is of insufficient sophistication to estimate dN/dS, and therefore recommends removal of this section. As a compromise, and given that some reviewers did want to see these effects explored in some way, we now describe the results more directly. We speak simply about the proportion of mutations that were non-synonymous, rather than about dN/dS. We also now focus our analysis on mutations within the spike gene. Finally, we have added binomial confidence intervals to each value provided to quantify uncertainty.

-- Figure 5C/5D brings up an very interesting pattern in Spike and should be discussed in greater detail. There clearly is more

non-synonymous and especially recurrent non-synonymous, substitutions in Spike for the MPV-associate branches. Why? Are these mutations occurring in preferred MPV contexts (TGT, TGC)? Is there enough time during a typical MPV-treatment to "fix" additional beneficial mutations following the introduction of MPV-induced errors? Is there any difference in Spike mutations between those sequences which transmit onward, and those which are dead ends? I suppose the "null" explanation is that sometimes (given the large number of replicates), MPV mutations just happen to hit the right targets in Spike. Yet, this could be tested by simulating mutations according to the predicted/observed distribution (conditioned on the number), and seeing how often you get recurrent or constellations of recurrent mutations.

We should be clear here that this figure does not present a comparison against untreated individuals. Mutations in untreated individuals are enriched for occurring in spike, and mutations in chronically infected individuals are especially enriched for being non-synonymous and occurring in spike. Indeed as our analysis of non-synonymous rates shows, the molnupiravir associated signature is associated with a lower proportion of non-synonymous mutations relative to other *long* branches. Rather, this figure shows that despite the fact that molnupiravir acts to induce random mutations, the mutations that we see in sequence databases are not wholly random but show biases in which non-synonymous mutations are more likely to occur in spike than surrounding regions, especially in the case of recurrent mutations. The explanation of this is that *selection*, by the immune system and other factors, is acting to enrich these mutations in consensus sequences. While in one sense that is unremarkable, the randomness of mutations identified is something that was highlighted in arguments in favour of molnupiravir's approval, and so these trends are important to highlight.

The idea of looking at the contexts of these Spike mutations is interesting, and we have revised the manuscript to present some information on this. We have expanded Fig 5D with a "context" column depicting the (reference-derived) context in which mutations occur.

This is as follows:

AA substitution	n	Mut. type	Context
S:P9L	12	C>T	CCA
S:D574N	12	G>A	TGA
S:E132K	7	G>A	TGA
S:E340K	7	G>A	TGA
S:A701V	7	C>T	GCA
S:A1070T	7	G>A	TGC
S:V1122M	7	G>A	TGT
S:K147E	6	A>G	CAA
S:A262T	6	G>A	TGC
S:G446S	6	G>A	TGG
S:V6I	5	G>A	TGT
S:G184S	5	G>A	GGG
S:G252S	5	G>A	TGG
S:G257S	5	G>A	AGG
S:V289I	5	G>A	TGT
S:R403K	5	G>A	AGA
S:K440R	5	A>G	AAT
S:R493Q	5	G>A	CGA
S:V595I	5	G>A	TGT
S:T1117I	5	C>T	ACA
S:V1128I	5	G>A	TGT

If one sees how many times each context occurs in this table, and annotates this onto a version of Fig S5C, that looks like this (Figure provided only for review):

So, as the reviewer predicts, TGT is the most common context, and more generally these mutations are far more likely to be placed above the diagonal line in these plots than below. This is consistent with the mutations in general truly being driven by molnupiravir rather than being advantageous mutations that happen to sneak through the high G-to-A threshold, and suggests that there is sometimes time for these mutations to be fixed following treatment. We also calculated the likelihood of observing contexts under multinomial models representing the molnupiravir spectrum and the BA.1 spectrum and found a dramatically increased likelihood of the molnupiravir spectrum. In the manuscript we now refer to the fact that TGT dominates in this list.

We agree that it would be interesting to understand whether the presence of particular mutations can predict whether a molnupiravir-mutated virus will persist in a patient and whether it will be transmitted. However we believe this will be best carried out after PANORAMIC sequence data is public and allows assessment of viral loads and the ability to culture virus for different molnupiravir-derived genotypes. We also believe the most convincing analyses will be those carried out by public health authorities who have data to definitively indicate onwards transmission in a greater number of cases.

-- Along those lines, one might expect spatial uniformity of MPV action, i.e. the fractions of mutations falling into a gene should be roughly proportional to its length corrected for mutation preference biases. Do we see this? If not, would indicate some sort of selection or deviation from assumptions.

We believe Fig 5C does a reasonable job of communicating this: synonymous mutations have a pretty flat distribution, as one would expect from spatial uniformity of mutagen action. Non-synonymous mutations diverge somewhat from this flat distribution, indicating a role for selection, which becomes clearer still when looking at the most recurrent mutations.

-- The R notebook in the attendant repository is, strangely, a PDF file. Makes reuse unnecessarily painful.

We thank the reviewer for pointing this out. While the PDF is useful in that it contains the plots, we have now added an additional raw quarto file to facilitate re-use. In the interests of reproducibility we have also expanded the "open data" version by uploaded pre-processed files to Zenodo that allow our open-data analyses to be reproduced. (These are distinct from the full dataset version due to GISAID's restrictions on data resharing).

Reviewer Reports on the Second Revision:

Referees' comments:

Referee #3 (Remarks to the Author):

I am satisfied with the revisions made by the authors; the remaining issues can be chalked up to a healthy difference of opinion and have no bearing on the suitability for publication.

I am happy to recommend this paper for acceptance and congratulate the authors on their tenacity and on making this important contribution.